# Groundwater storage dynamics in the world's large aquifer systems from GRACE: uncertainty and role of extreme precipitation

Mohammad Shamsudduha[1,2,*] and Richard G. Taylor[1]

[1] Department of Geography, University College London, London, UK

[2] Department of Geography, University of Sussex, Falmer, Brighton, UK

[*] Corresponding author: M. Shamsudduha (M.Shamsudduha@sussex.ac.uk)

## Abstract

Under variable and changing climates groundwater storage sustains vital ecosystems and enables freshwater withdrawals globally for agriculture, drinking-water, and industry. Here, we assess recent changes in groundwater storage ($\Delta$GWS) from 2002 to 2016 in 37 of the world's large aquifer systems using an ensemble of datasets from the Gravity Recovery and Climate Experiment (GRACE) and Land Surface Models (LSMs). Ensemble GRACE-derived $\Delta$GWS is well reconciled to in-situ observations ($r = 0.62$–$0.86$, $p$ value $<0.001$) for two tropical basins with regional piezometric networks and contrasting climate regimes. Trends in GRACE-derived $\Delta$GWS are overwhelmingly non-linear; indeed, linear declining trends adequately ($R^2 > 0.5$, $p$ value $<0.001$) explain variability in only two aquifer systems. Non-linearity in $\Delta$GWS derives, in part, from the episodic nature of groundwater replenishment associated with extreme annual ($>90^{th}$ percentile, 1901–2016) precipitation and is inconsistent with prevailing narratives of global-scale groundwater depletion at the scale of GRACE footprint ($\sim$200,000 km$^2$). Substantial uncertainty remains in estimates of GRACE-derived $\Delta$GWS, evident from 20 realisations presented here, but these data provide a regional context to changes in groundwater storage observed more locally through piezometry.

## 1   Introduction

Groundwater is estimated to account for between a quarter and a third of the world's annual freshwater withdrawals to meet agricultural, industrial and domestic demand (Döll et al., 2012; Wada et al., 2014; Hanasaki et al., 2018). As the world's largest distributed store of freshwater, groundwater plays a vital role in sustaining ecosystems and enabling adaptation to increased variability in rainfall and river discharge brought about by climate change (Taylor et al., 2013a). Sustained reductions in the volume of groundwater (i.e. groundwater depletion) resulting from human withdrawals or changes in climate have historically been observed as declining groundwater levels recorded in wells (Scanlon et al., 2012a; Castellazzi et al., 2016; MacDonald et al., 2016). The limited distribution and duration of piezometric records hinder, however, direct observation of changes in groundwater storage globally including many of the world's large aquifer systems (WHYMAP and Margat, 2008).

Since 2002 the Gravity Recovery and Climate Experiment (GRACE) has enabled large-scale ($\geq$ 200,000 km$^2$) satellite monitoring of changes in total terrestrial water storage ($\Delta$TWS) globally (Tapley et al., 2004). As the twin GRACE satellites circle the globe ~15 times a day they measure the inter-satellite distance at a minute precision (within one micron) and provide $\Delta$TWS for the entire earth approximately every 30 days. GRACE satellites sense movement of total terrestrial water mass derived from both natural (e.g. droughts) and anthropogenic (e.g. irrigation) influences globally (Rodell et al., 2018). Changes in groundwater storage (GRACE-derived $\Delta$GWS) are computed from $\Delta$TWS after deducting contributions (equation 1) that arise from other terrestrial water stores including soil moisture storage ($\Delta$SMS), surface water storage ($\Delta$SWS), and the snow water storage ($\Delta$SNS) using data from Land Surface Models (LSMs) either exclusively (Rodell et al., 2009; Famiglietti et al., 2011; Scanlon et al., 2012a; Famiglietti and Rodell, 2013; Richey et al., 2015; Thomas et

al., 2017) or in combination with in situ observations (Rodell et al., 2007; Swenson et al.,
2008; Shamsudduha et al., 2012).
$\Delta GWS = \Delta TWS - (\Delta SMS + \Delta SWS + \Delta SNS)$ (1)
Substantial uncertainty persists in the quantification of changes in terrestrial water stores
from GRACE measurements that are limited in duration (2002 to 2016), and the application
of uncalibrated, global-scale LSMs (Shamsudduha et al., 2012; Döll et al., 2014; Scanlon et
al., 2018). Computation of $\Delta GWS$ from GRACE $\Delta TWS$ is argued, nevertheless, to provide
evaluations of large-scale changes in groundwater storage where regional-scale piezometric
networks do not currently exist (Famiglietti, 2014).
Previous assessments of changes in groundwater storage using GRACE in the world's 37
large aquifer systems (Richey et al., 2015; Thomas et al., 2017) (Fig. 1, Table 1) have raised
concerns about the sustainability of human use of groundwater resources. One analysis
(Richey et al., 2015) employed a single GRACE $\Delta TWS$ product (CSR) in which changes in
subsurface storage ($\Delta SMS + \Delta GWS$) were attributed to $\Delta GWS$. This study applied linear
trends without regard to their significance to compute values of GRACE-derived $\Delta GWS$ over
11 years from 2003 to 2013, and concluded that the majority of the world's aquifer systems
($n = 21$) are either "overstressed" or "variably stressed". A subsequent analysis (Thomas et
al., 2017) employed a different GRACE $\Delta TWS$ product (Mascons) and estimated $\Delta SWS$
from LSM data for both surface and subsurface runoff, though the latter is normally
considered to be groundwater recharge (Rodell et al., 2004). Using performance metrics
normally applied to surface water systems including dams, this latter analysis classified
nearly a third ($n = 11$) of the world's aquifer systems as having their lowest sustainability
criterion.
Here, we update and extend the analysis of ΔGWS in the world's 37 large aquifer systems
using an ensemble of three GRACE ΔTWS products (CSR, Mascons, GRGS) over a 14-year
period from August 2002 to July 2016. To isolate GRACE-derived ΔGWS from GRACE
ΔTWS, we employ estimates of ΔSMS, ΔSWS and ΔSNS from five LSMs (CLM, Noah,
VIC, Mosaic, Noah v.2.1) run by NASA's Global Land Data Assimilation System (GLDAS).
As such, we explicitly account for the contribution of ΔSWS to ΔTWS, which has been
commonly overlooked (Rodell et al., 2009; Richey et al., 2015; Bhanja et al., 2016) despite
evidence of its significant contribution to ΔTWS (Kim et al., 2009; Shamsudduha et al.,
2012; Getirana et al., 2017). Further, we characterise trends in time-series records of
GRACE-derived ΔGWS by employing a non-parametric, Seasonal-Trend decomposition
procedure based on Loess (STL) (Cleveland et al., 1990) that allows for resolution of
seasonal, trend and irregular components of GRACE-derived ΔGWS for each large aquifer
system. In contrast to linear or multiple-linear regression-based techniques, STL assumes
neither that data are normally distributed nor that the underlying trend is linear
(Shamsudduha et al., 2009; Humphrey et al., 2016; Sun et al., 2017).

## 88   2    Data and Methods

### 89   2.1    Global large aquifer systems

We use the World-wide Hydrogeological Mapping and Assessment Programme (WHYMAP)
Geographic Information System (GIS) dataset for the delineation of world's 37 Large Aquifer
Systems  (Fig. 1, Table1) (WHYMAP and Margat, 2008). The WHYMAP network, led by
the German Federal Institute for Geosciences and Natural Resources (BGR), serves as a
central repository and hub for global groundwater data, information, and mapping with a goal
of assisting regional, national, and international efforts toward sustainable groundwater

management (Richts et al., 2011). The largest aquifer system in this dataset (Supplementary

Table S1) is the East European Aquifer System (WHYMAP no. 33; area: 2.9 million km$^2$)

and the smallest one the California Central Valley Aquifer System (WHYMAP no. 16; area:

71,430 km$^2$), which is smaller than the typical sensing area of GRACE (~200,000 km$^2$).

However, Longuevergne et al. (2013) argue that GRACE satellites are sensitive to total mass

changes at a basin scale so ΔTWS measurements can be applied to smaller basins if the

magnitude of temporal mass changes is substantial due to mass water withdrawals (e.g.,

intensive groundwater-fed irrigation). Mean and median sizes of these large aquifers are

~945,000 km$^2$ and ~600,000 km$^2$, respectively.

## 2.2   GRACE products

We use post-processed, gridded (1° × 1°) monthly GRACE TWS data from CSR land

(Landerer and Swenson, 2012) and JPL Global Mascon (Watkins et al., 2015; Wiese et al.,

2016) solutions from NASA's dissemination site (http://grace.jpl.nasa.gov/data), and a third

GRGS GRACE solution (CNES/GRGS release RL03-v1) (Biancale et al., 2006) from the

French Government space agency, Centre National D'études Spatiales (CNES). To address

the uncertainty associated with different GRACE processing strategies (CSR, JPL-Mascons,

GRGS), we apply an ensemble mean of the three GRACE solutions (Bonsor et al., 2018).

CSR land solution (version RL05.DSTvSCS1409) is post-processed from spherical

harmonics released by the Centre for Space Research (CSR) at the University of Texas at

Austin. CSR gridded datasets are available at a monthly timestep and a spatial resolution of

1° × 1° (~111 km at equator) though the actual spatial resolution of GRACE footprint

(Scanlon et al., 2012a) is 450 km × 450 km or ~200,000 km$^2$. To amplify TWS signals we

apply the dimensionless scaling factors provided as 1° × 1° bins that are derived from

minimising differences between TWS estimated from GRACE and the hydrological fields

from the Community Land Model (CLM4.0) (Landerer and Swenson, 2012). JPL-Mascons
(version RL05M_1.MSCNv01) data processing involves the same glacial isostatic adjustment
correction but applies no spatial filtering as JPL-RL05M directly relates inter-satellite range-
rate data to mass concentration blocks (Mascons) to estimate monthly gravity fields in terms
of equal area 3° × 3° mass concentration functions in order to minimise measurement errors.
Gridded mascon fields are provided at a spatial sampling of 0.5° in both latitude and
longitude (~56 km at the equator). Similar to CSR product, dimensionless scaling factors are
provided as 0.5° × 0.5° bins (Shamsudduha et al., 2017) to apply to the JPL-Mascons product
that also derive from the Community Land Model (CLM4.0) (Wiese et al., 2016). The scaling
factors are multiplicative coefficients that minimize the difference between the smoothed and
unfiltered monthly ΔTWS variations from the CLM4.0 hydrology model (Wiese et al., 2016).
Finally, GRGS GRACE (version RL03-v1) monthly gridded solutions of a spatial resolution
of 1° × 1° are extracted and aggregated time-series data are generated for each aquifer
system. A description of the estimation method of ΔGWS from GRACE and in-situ
observations is provided below.
**2.3    Estimation of ΔGWS from GRACE**
We apply monthly measurements of terrestrial water storage anomalies (ΔTWS) from
Gravity Recovery and Climate Experiment (GRACE) satellites, and simulated records of soil
moisture storage (ΔSMS), surface runoff or surface water storage (ΔSWS) and snow water
equivalent (ΔSNS) from NASA's Global Land Data Assimilation System (GLDAS version
1.0) at 1° × 1° grids for the period of August 2002 to July 2016 to estimate (equation 1)
groundwater storage changes (ΔGWS) in the 37 WHYMAP large aquifer systems. This
approach is consistent with previous global (Thomas et al., 2017) and basin-scale (Rodell et
al., 2009; Asoka et al., 2017; Feng et al., 2018) analyses of ΔGWS from GRACE. We apply 3
gridded GRACE products (CSR, JPL-Mascons, GRGS) and an ensemble mean of ΔTWS and
individual storage component of ΔSMS and ΔSWS from 4 Land Surface Models (LSMs:
CLM, Noah, VIC, Mosaic), and a single ΔSNS from Noah model (GLDAS version 2.1) to
derive a total of 20 realisations of ΔGWS (Table S5) for each of the 37 aquifer systems. We
then averaged all the GRACE-derived ΔGWS estimates to generate an ensemble mean
ΔGWS time-series record for each aquifer system. GRACE and GLDAS LSMs derived
datasets are processed and analysed in R programming language (R Core Team, 2017).
**2.4   GLDAS Land Surface Models**
To estimate GRACE-derived ΔGWS using equation (1), we use simulated soil moisture
storage (ΔSMS), surface runoff, as a proxy for surface water storage ΔSWS (Getirana et al.,
2017; Thomas et al., 2017), and snow water equivalent (ΔSNS) from NASA's Global Land
Data Assimilation System (GLDAS). GLDAS system (https://ldas.gsfc.nasa.gov/gldas/)
drives multiple, offline (not coupled to the atmosphere) Land Surface Models globally
(Rodell et al., 2004), at variable grid resolutions (from 2.5° to 1 km), enabled by the Land
Information System (LIS) (Kumar et al., 2006). Currently, GLDAS (version 1) drives four
land surface models (LSMs): Mosaic, Noah, the Community Land Model (CLM), and the
Variable Infiltration Capacity (VIC). We apply monthly ΔSMS (sum of all soil profiles) and
ΔSWS data at a spatial resolution of 1° × 1° from 4 GLDAS LSMs: the Community Land
Model (CLM, version 2.0) (Dai et al., 2003), Noah (version 2.7.1) (Ek et al., 2003), the
Variable Infiltration Capacity (VIC) model (version 1.0) (Liang et al., 2003), and Mosaic
(version 1.0) (Koster and Suarez, 1992). The respective total depths of modelled soil profiles
are 3.4 m, 2.0 m, 1.9 m and 3.5 m in CLM (10 vertical layers), Noah (4 vertical layers), VIC
(3 vertical layers), and Mosaic (3 vertical layers) (Rodell et al., 2004). For snow water
equivalent (ΔSNS), we use simulated data from Noah (v.2.1) model (GLDAS version 2.1)
that is forced by the global meteorological data set from Princeton University (Sheffield et
al., 2006); LSMs under GLDAS (version 1) are forced by the CPC Merged Analysis of
Precipitation (CMAP) data (Rodell et al., 2004).

## 2.5 Global precipitation datasets

To evaluate the relationships between precipitation and GRACE-derived ΔGWS, we use a
high-resolution (0.5 degree) gridded, global precipitation dataset (version 4.01) (Harris et al.,
2014) available from the Climatic Research Unit (CRU) at the University of East Anglia
(https://crudata.uea.ac.uk/cru/data/hrg/). In light of uncertainty in observed precipitation
datasets globally, we test the robustness of relationship between precipitation and
groundwater storage using the GPCC (Global Precipitation Climatology Centre) precipitation
dataset (Schneider et al., 2017) (https://www.esrl.noaa.gov/psd/data/gridded/data.gpcc.html)
from 1901 to 2016. Time-series (January 1901 to July 2016) of monthly precipitation from
CRU and GPCC datasets for the WHYMAP aquifer systems were analysed and processed in
R programming language (R Core Team, 2017).

## 2.6 Seasonal-Trend Decomposition (STL) of GRACE ΔGWS

Monthly time-series records (Aug 2002 to Jul 2016; supplementary Figs. S1-S36) of the
ensemble mean GRACE ΔTWS and GRACE-derived ΔGWS were decomposed to seasonal,
trend and remainder or residual components using a non-parametric time series
decomposition technique known as "Seasonal-Trend decomposition procedure based on a
locally weighted regression method called Loess (STL)" (Cleveland et al., 1990). Loess is a
nonparametric method so that the fitted curve is obtained empirically without assuming the
specific nature of any structure that may exist within the data (Jacoby, 2000). A key
advantage of STL method is that it reveals relatively complex structures in time-series data
that could easily be overlooked using traditional statistical methods such as linear regression.
STL decomposition technique has previously been used to analyse GRACE ΔTWS regionally
(Hassan and Jin, 2014) and globally (Humphrey et al., 2016). GRACE-derived ΔGWS time-
series records for each aquifer system were decomposed using the STL method (see equation
2) in the R programming language (R Core Team, 2017) as:
$Y_t = T_t + S_t + R_t$                                                                (2)
where $Y_t$ is the monthly ΔGWS at time $t$, $T_t$ is the trend component; $S_t$ is the seasonal
component; and $R_t$ is a remainder (residual or irregular) component.
The STL method consists of a series of smoothing operations with different moving window
widths chosen to extract different frequencies within a time series, and can be regarded as an
extension of classical methods for decomposing a series into its individual components
(Chatfield, 2003). The nonparametric nature of the STL decomposition technique enables
detection of nonlinear patterns in long-term trends that cannot be assessed through linear
trend analyses (Shamsudduha et al., 2009). For STL decomposition, it is necessary to choose
values of smoothing parameters to extract trend and seasonal components. Selection of
parameters in STL decomposition is a subjective process. The choice of the seasonal
smoothing parameter determines the extent to which the extracted seasonal component varies
from year to year: a large value will lead to similar components in all years whereas a small
value will allow the extracted component to track the observations more closely. Similar
comments apply to the choice of smoothing parameter for the trend component. We
experimented with several different choices of smoothing parameters (see supplementary Fig.
S37) and checked the residuals (i.e. remainder component) for the overall performance of the
STL decomposition model. We conducted the Shapiro-Wilk normality test on the residuals
after fitting the STL smooth line with a range of trend-cycle (*t.window*) and seasonal
(*s.window*) windows and compared the *p* values. Visualization of the results with several
smoothing parameters (supplementary Fig. S37) and the corresponding smaller *p* values (i.e.,
*p* value <0.01) of the normality test suggested that the overall structure of time series at all
sites could be captured reasonably well using window widths of 13 for the seasonal
component and 37 for the trend. We apply the STL decomposition with a robust fitting of the
loess smoother (Cleveland et al., 1990) to ensure that the fitting of the curvilinear trend does
not have an adverse effect due to extreme outliers in the time-series data (Jacoby, 2000).
Finally, to make the interpretation and comparison of nonlinear trends across all 37 aquifer
systems, smoothing parameters were then fixed for all subsequent STL analyses.

## 225     3     Results

### 226     3.1     Variability in ΔTWS of the large aquifer systems

Ensemble mean time series of GRACE ΔTWS for the world's 37 large aquifer systems are
shown in Fig. 2 (High Plains Aquifer System, no. 17) and supplementary Figs. S1-S36
(remaining 36 aquifer systems). The STL decomposition of an ensemble GRACE ΔTWS in
the High Plains Aquifer System (no. 17) decomposes the time series into seasonal, trend and
residual components (see supplementary Fig. S37). Variance (square of the standard
deviation) in monthly GRACE ΔTWS (Figs. 3a and 4, Supplementary Table S1) is highest
(>100 cm$^2$) primarily under monsoonal precipitation regimes within the Inter-Tropical
Convergence Zone (e.g. Upper Kalahari-Cuvelai-Zambezi-11, Amazon-19, Maranho-20,
Ganges-Brahmaputra-24). The sum of individual components derived from the STL
decomposition (i.e., seasonal, trend and irregular or residual) approximates the overall
variance in time-series data. The majority of the variance (>50%) in ΔTWS is explained by
seasonality (Fig. 3a); non-linear (curvilinear) trends represent <25% of the variance in ΔTWS
with the exception of the Upper Kalahari-Cuvelai-Zambezi-11 (42%). In contrast, variance in
GRACE ΔTWS in most hyper-arid and arid basins is low (Fig. 3a), <10 cm$^2$ (e.g., Nubian-1,
NW Sahara-2, Murzuk-Djado-3, Taodeni-Tanezrouft-4, Ogaden-Juba-9, Lower Kalahari-
Stampriet-12, Karoo-13, Tarim-31) and largely (> 65%) attributed to ΔGWS (Supplementary
Table S2). Overall, changes in ΔTWS (i.e., difference between two consecutive hydrological
years) are correlated (Pearson correlation, $r >0.5$, $p$ value <0.01) to annual precipitation for
25 of the 37 large aquifer systems (Table S1). GRACE ΔTWS in aquifer systems under
monsoonal precipitation regimes is strongly correlated to rainfall with a lag of 2 months ($r$
>0.65, $p$ value <0.01).
**3.2 GRACE-ΔGWS and evidence from in-situ piezometry**
Evaluations of computed GRACE-derived ΔGWS using in situ observations are limited
spatially and temporally by the availability of piezometric records (Swenson et al., 2006;
Strassberg et al., 2009; Scanlon et al., 2012b; Shamsudduha et al., 2012; Panda and Wahr,
2015; Feng et al., 2018). Consequently, comparisons of GRACE and in situ ΔGWS remain
opportunity-driven and, here, comprise the Limpopo Basin in South Africa and Bengal Basin
in Bangladesh where we possess time series records of adequate duration and density. The
Bengal Basin is a part of the Ganges-Brahmaputra aquifer system (aquifer no. 24) whereas
the Limpopo Basin is located between the Lower Kalahari-Stampriet Basin (aquifer no. 12)
and the Karoo Basin (aquifer no. 13). The two basins feature contrasting climates (i.e.
tropical humid versus tropical semi-arid) and geologies (i.e. unconsolidated sands versus
weathered crystalline rock) that represent key controls on the magnitude and variability
expected in ΔGWS. Both basins are in the tropics and, as such, serve less well to test the
computation of GRACE-derived ΔGWS at mid and high latitudes.
In the Bengal Basin, computed GRACE and in situ ΔGWS demonstrate an exceptionally
strong seasonal signal associated with monsoonal recharge that is amplified by dry-season
abstraction (Shamsudduha et al., 2009; Shamsudduha et al., 2012) and high storage of the
regional unconsolidated sand aquifer, represented by a bulk specific yield ($S_y$) of 10% (Fig.
S38a). Time-series of GRACE and LSMs are shown in Fig. S39. The ensemble mean time
series of computed GRACE ΔGWS from three GRACE TWS solutions and five NASA
GLDAS LSMs is strongly correlated ($r = 0.86$, $p$ value <0.001) to in situ ΔGWS derived
from a network of 236 piezometers (mean density of 1 piezometer per 610 km$^2$) for the
period of 2003 to 2014. In the semi-arid Limpopo Basin where mean annual rainfall (469 mm
for the period of 2003 to 2015) is one-fifth of that in the Bengal Basin (2,276 mm), the
seasonal signal in ΔGWS, primarily in weathered crystalline rocks with a bulk $S_y$ of 2.5%, is
smaller (Fig. S38b). Time-series of GRACE and LSMs are shown in Fig. S40. Comparison of
in situ ΔGWS, derived from a network of 40 piezometers (mean density of 1 piezometer per
1,175 km$^2$), and computed GRACE-derived ΔGWS shows broad correspondence ($r = 0.62$, $p$
value <0.001) though GRACE-derived ΔGWS is 'noisier'; intra-annual variability may result
from uncertainty in the representation of other terrestrial stores using LSMs that are used to
compute GRACE-derived ΔGWS from GRACE ΔTWS. The magnitude of uncertainty in
monthly ΔSWS, ΔSMS, and ΔSNS that are estimated by GLDAS LSMs to compute
GRACE-derived ΔGWS in each large-scale aquifer system, is depicted in Fig. 2 and
supplementary Figs. S1-S36. The favourable, statistically significant correlations between the
computed ensemble mean GRACE-derived ΔGWS and in situ ΔGWS shown in these two
contrasting basins indicate that, at large scales (~200,000 km$^2$), the methodology used to
compute GRACE-derived ΔGWS has merit.
**3.3    Trends in GRACE-ΔGWS time series**
Computation of GRACE-derived ΔGWS for the 37 large-scale aquifers globally is shown in
Figs. 2 and 5. Figure 2 shows the ensemble GRACE ΔTWS and GLDAS LSM datasets used
to compute GRACE-derived ΔGWS for the High Plains Aquifer System in the USA (aquifer
no. 17 in Fig. 1); datasets used for all other large-scale aquifer systems are given in the
Supplementary Material (Figs. S1–S36).  In addition to the ensemble mean, we show
uncertainty in GRACE-derived ΔGWS associated with 20 realisations from GRACE products
and LSMs. Monthly time-series data of ensemble GRACE-derived ΔGWS for the other 36
large-scale aquifers are plotted (absolute scale) in Fig. 5 (in black) and fitted with a Loess-
based trend (in blue). For all but five large aquifer systems (e.g., Lake Chad Basin-
WHYMAP no. 7, Umm Ruwaba-8, Amazon-19, West Siberian Basin-25, and East European-
33), the dominant time-series component explaining variance in GRACE-derived ΔGWS is
trend (Fig. 3b, and supplementary Figs. S41-S77). Trends in GRACE-derived ΔGWS are,
however, overwhelmingly non-linear (curvilinear); linear trends adequately ($R^2 > 0.5$, $p$ value
<0.05) explain variability in GRACE-derived ΔGWS in just 5 of 37 large-scale aquifer
systems and of these, only two (Arabian-22, Canning-37) are declining. GRACE-derived
ΔGWS for three intensively developed, large-scale aquifer systems (Supplementary Table S1:
California Central Valley-16, Ganges-Brahmaputra-24, North China Plains-29) show
episodic declines (Fig. 5) though, in each case, their overall trend from 2002 to 2016 is
declining but non-linear (Fig. 1).
**3.4   Computational uncertainty in GRACE-ΔGWS**
For several large aquifer systems primarily in arid and semi-arid environments, we identify
anomalously negative or positive estimates of GRACE-derived ΔGWS that deviate
substantially from underlying trends (Fig. 6 and supplementary Fig. S78). For example, the
semi-arid Upper Kalahari-Cuvelai-Zambezi Basin (11) features an extreme, negative anomaly
in GRACE-derived ΔGWS (Fig. 6a) in 2007-08 that is the consequence of simulated values
of terrestrial stores (ΔSWS + ΔSMS) by GLDAS LSMs that exceed the ensemble GRACE
ΔTWS signal. Inspection of individual time-series data for this basin (Fig. S11) reveals
greater consistency in the three GRACE-$\Delta$TWS time-series data (variance of CSR: 111 cm$^2$;
Mascons: 164 cm$^2$; GRGS: 169 cm$^2$) compared to simulated $\Delta$SMS among the 4 GLDAS
LSMs (variance of CLM: 9 cm$^2$; Mosaic: 90 cm$^2$; Noah: 98 cm$^2$; VIC is 110 cm$^2$). In the
humid Congo Basin (10), positive $\Delta$TWS values in 2006-07 but negative $\Delta$SMS values
produce anomalously high values of GRACE-derived $\Delta$GWS (Fig. 6b, Fig. S10). In the
snow-dominated, humid Angara-Lena Basin (27), a strongly positive, combined signal of
$\Delta$SNS + $\Delta$SWS exceeding $\Delta$TWS leads to a very negative estimation of $\Delta$GWS when
groundwater is following a rising trend (Fig. 6c, Fig. S26).
**3.5  GRACE $\Delta$GWS and extreme precipitation**
Non-linear trends in GRACE-derived $\Delta$GWS (i.e., difference in STL trend component
between two consecutive years) demonstrate a significant association with precipitation
anomalies from CRU dataset for each hydrological year (i.e., percent deviations from mean
annual precipitation between 2002 and 2016) in semi-arid environments (Fig. 7, Pearson
correlation, $r = 0.62$, $p < 0.001$). These associations over extreme hydrological years are
particularly strong in a number of individual aquifer systems (Fig. 5; Supplementary Tables
S3 and S4) including the Great Artesian Basin (36) ($r = 0.93$), California Central Valley (16)
($r = 0.88$), North Caucasus Basin (34) ($r = 0.65$), Umm Ruwaba Basin (8) ($r = 0.64$), and
Ogalalla (High Plains) Aquifer (17) ($r = 0.64$). In arid aquifer systems, overall associations
between GRACE $\Delta$GWS and precipitation anomalies are statistically significant but
moderate ($r = 0.36$, $p < 0.001$); a strong association is found only for the Canning Basin (37)
($r = 0.52$).  In humid (and sub-humid) aquifer systems, no overall statistically significant
association is found yet strong correlations are noted for two temperate aquifer systems
(Northern Great Plains Aquifer (14), $r = 0.51$; Angara−Lena Basin (27), $r = 0.54$); weak
correlations are observed in the humid tropics for the Maranhao Basin (20, $r = 0.24$) and
Ganges-Brahmaputra Basin (24, $r = 0.28$).
Distinct rises observed in GRACE-derived ΔGWS correspond with extreme seasonal
(annual) precipitation (Fig. 5; Table S3 and Table S4). In the semi-arid Great Artesian Basin
(aquifer no. 36) (Fig. 5 and supplementary Fig. S35), two consecutive years (2009–10 and
2010–11) of statistically extreme (i.e., >90th percentile, period: 1901 to 2016) monthly
precipitation interrupt a multi-annual (2002 to 2009) declining trend. Pronounced rises in
GRACE-derived ΔGWS in response to extreme annual rainfall are visible in other semi-arid,
large aquifer systems including the Umm Ruwaba Basin (8) in 2007, Lower Kalahari-
Stampriet Basin (12) in 2011, California Central Valley (16) in 2005, Ogalalla (High Plains)
Aquifer (17) in 2015, and Indus Basin (23) in 2010 and 2015 (Tables S3 and S4 and Figs. S2,
S8, S12, S16, S22). Similar rises in GRACE-derived ΔGWS in response to extreme annual
rainfall in arid basins include the Lake Chad Basin (7) in 2012 and Ogaden-Juba Basin (9) in
2013 (Table S3 and Figs. S7, S9). In the Canning Basin, a substantial rise in GRACE-derived
ΔGWS occurs in 2010–11 (Tables S3 and S4 and Fig. S36) in response to extreme annual
rainfall though the overall trend is declining.
Non-linear trends that feature substantial rises in GRACE-derived ΔGWS in response to
extreme annual precipitation under humid climates, are observed in the Maranhao Basin (20)
in 2008-09, Guarani Aquifer System (21) in 2015-16, and North China Plains (29) in 2003.
Consecutive years of extreme precipitation in 2012 and 2013 also generate a distinct rise in
GRACE-derived ΔGWS in the Song-Liao Plain (30) (Tables S3 and S4 and Figs. S29). In the
heavily developed (Table S2) Ganges-Brahmaputra Basin (24), a multi-annual (2002 to 2010)
declining trend is halted by an extreme (i.e., highest over the GRACE period of 2002 to 2016
but 59th percentile over the period of 1901 to 2016 using CRU dataset) annual precipitation in
2011 (Tables S3 and S4 and Figs. S23). Consecutive years from 2014 to 2015 of extreme
annual precipitation increase GRACE-derived ΔGWS and disrupt a multi-annual declining
trend in the West Siberian Artesian Basin (25) (Tables S3 and S4 and Figs. S24). In the sub-
humid Northern Great Plains (14), distinct rises in GRACE-derived ∆GWS occur in 2010
(Tables S3 and S4 and Figs. S14) in response to extreme annual precipitation though the
overall trend is linear and rising. The overall agreement in mean annual precipitation between
the CRU and GPCC datasets for the period of 1901 to 2016 is strong (median correlation
coefficient in 37 aquifer systems, $r = 0.92$).

## 369   4    Discussion

### 370   4.1    Uncertainty in GRACE-derived ∆GWS

We compute the range of uncertainty in GRACE-derived ∆GWS associated with 20 potential
realisations from applied GRACE (CSR, JPL-Mascons, GRGS) products and LSMs (CLM,
Noah, VIC, Mosaic). Uncertainty is generally higher for aquifers systems located in arid to
hyper-arid environments (Table 2, see supplementary Fig. S79). Computation of GRACE-
derived ∆GWS relies upon uncalibrated simulations of individual terrestrial water stores (i.e.,
∆SWS, ∆SWS, ∆SNS) from LSMs to estimate ∆GWS from GRACE ∆TWS. A recent
global-scale comparison of ∆TWS estimated by GLDAS LSMs and GRACE (Scanlon et al.,
2018) indicates that LSMs systematically underestimate water storage changes. Further, the
absence of river-routing and representation of lakes and reservoirs in the estimation of ∆SWS
by LSMs constrains computation of GRACE ∆GWS as similarly recognised by Scanlon et al.
(2019). Finally, substantial variability in ∆SMS among GLDAS models and the limited depth
(<3.5 m below ground level) to the deepest soil layer over which these LSMs simulate ∆SMS
also hamper estimation of GRACE ∆GWS, especially in drylands where the thickness of
unsaturated zones may an order of magnitude greater (Scanlon et al., 2009).
We detect probable errors in GLDAS LSM data from events that produce large deviations in
GWS (Fig. 5). These errors occur because GRACE-derived ∆GWS is computed as residual
(equation 1); overestimation (or underestimation) of these combined stores produces negative
(or positive) values of GRACE-derived ΔGWS when the aggregated value of other terrestrial
water stores is strongly positive (or negative) and no lag is assumed (Shamsudduha et al.,
2017). Evidence from limited piezometric data presented here and elsewhere (Panda and
Wahr, 2015; Feng et al., 2018) suggests that the dynamics in computed GRACE-derived
ΔGWS are nonetheless reasonable yet the amplitude in ΔGWS from piezometry is scalable
due to uncertainty in the applied $S_y$ (Shamsudduha et al., 2012).
Assessments of ΔGWS derived from GRACE are constrained by both their limited timespan
(2002–2016) and course spatial resolution (>200,000 km$^2$). For example, centennial-scale
piezometry in the Ganges-Brahmaputra aquifer system (no. 24) reveals that recent
groundwater depletion, (i.e., groundwater withdrawals that are unlikely to be replenished
within a century as per Bierkens and Wada (2019)), in NW India traced by GRACE (Fig. 5
and supplementary Fig. S23) (Rodell et al., 2009; Chen et al., 2014) follows more than a
century of groundwater accumulation (see supplementary Fig. S80) through leakage of
surface water via a canal network constructed primarily during the 19$^{th}$ century (MacDonald
et al., 2016). Long-term piezometric records from central Tanzania and the Limpopo Basin of
South Africa (Supplementary Fig. S81) show dramatic increases in ΔGWS associated with
extreme seasonal rainfall events that occurred prior to 2002 and thus provide a vital context
to the more recent period of ΔGWS estimated by GRACE. At regional scales, GRACE-
derived ΔGWS can differ substantially from more localised, in situ observations of ΔGWS
from piezometry. In the Karoo Basin (aquifer no. 13), GRACE-derived ΔGWS is also rising
(Fig. 5 and supplementary Fig. S13) over periods during which groundwater depletion has
been reported in parts of the basin (Rosewarne et al., 2013). In the Guarani Aquifer System
(21), groundwater depletion is reported from 2005 to 2009 in Ribeiro Preto near Sao Paulo as
a result of intensive groundwater withdrawals for urban water supplies and irrigation of
sugarcane (Foster et al., 2009) yet GRACE-derived ∆GWS over this same period is rising.

### 4.2 Variability in GRACE ∆GWS and role of extreme precipitation

Non-linear trends in GRACE-derived ∆GWS arise, in part, from inter-annual variability in
precipitation which has similarly been observed in analyses of GRACE ∆TWS (Humphrey et
al., 2016; Sun et al., 2017; Bonsor et al., 2018). Annual precipitation in the Great Artesian
Basin (aquifer no. 36) provides a dramatic example of how years (2009–10, 2010–11 from
both CRU and GPCC datasets) of extreme precipitation can generate anomalously high
groundwater recharge that arrests a multi-annual declining trend (Fig. 5), increasing
variability in GRACE-derived ∆GWS over the relatively short period (15 years) of GRACE
data. The disproportionate contribution of episodic, extreme rainfall to groundwater recharge
has previously been shown by (Taylor et al., 2013b) from long-term piezometry in semi-arid
central Tanzania where nearly 20% of the recharge observed over a 55-year period resulted
from a single season of extreme rainfall, associated with the strongest El Niño event (1997–
1998) of the last century (Supplementary Fig. S81a). Further analysis from multi-decadal
piezometric records in drylands across tropical Africa (Cuthbert et al., 2019) confirm this bias
in response to intensive precipitation.
The dependence of groundwater replenishment on extreme annual precipitation indicated by
GRACE-derived ∆GWS for many of the world's large aquifer systems is consistent with
evidence from other sources. In a pan-tropical comparison of stable-isotope ratios of oxygen
($^{18}O$:$^{16}O$) and hydrogen ($^{2}H$:$^{1}H$) in rainfall and groundwater, Jasechko and Taylor (2015)
show that recharge is biased to intensive monthly rainfall, commonly exceeding the 70[th]
percentile. In humid Uganda, Owor et al. (2009) demonstrate that groundwater recharge
observed from piezometry is more strongly correlated to daily rainfall exceeding a threshold
(10 mm) than all daily rainfalls. Periodicity in groundwater storage indicated by both
GRACE and in situ data has been associated with large-scale synoptic controls on
precipitation (e.g., El Niño Southern Oscillation, Pacific Decadal Oscillation,) in southern
Africa (Kolusu et al., 2019), and have been shown to amplify recharge in major US aquifers
(Kuss and Gurdak, 2014) and groundwater depletion in India (Mishra et al., 2016).
In some large-scale aquifer systems, GRACE-derived ΔGWS exhibits comparatively weak
correlations to precipitation. In the semi-arid Iullemmeden-Irhazer Aquifer (6) variance in
rainfall over the period of GRACE observation following the multi-decadal Sahelian drought
is low (Table S1) and the net rise in GRACE-derived ΔGWS is associated with changes in
the terrestrial water balance resulting from land-cover change (Ibrahim et al., 2014). In the
Amazon (16), rising trends in GRACE-derived ΔGWS, which are aligned to ΔTWS reported
previously by Scanlon et al. (2018) and Rodell et al. (2018), occur during a period (2010–
2016; see supplementary Table S18) that is the driest since the 1980s (Chaudhari et al.,
2019); analyses over the longer term (1980–2015) point nevertheless to an overall
intensification of the Amazonian hydrological cycle.
**4.3 Trends in GRACE ΔGWS under global change**
Our analysis identifies non-linear trends in GRACE-derived ΔGWS for the vast majority (32
of 37) of the world's large aquifer systems (Figs. 1, 5 and 8). Non-linearity reflects, in part,
the variable nature of groundwater replenishment observed at the scale of the GRACE
footprint that is consistent with more localised, emerging evidence from multi-decadal
piezometric records (Taylor et al., 2013b) (Supplementary Fig. S81a). The variable and often
episodic nature of groundwater replenishment complicates assessments of the sustainability
of groundwater withdrawals and highlights the importance of long-term observations over
decadal timescales in undertaking such evaluations. Dramatic rises in freshwater withdrawals,
primarily associated with the expansion of irrigated agriculture in semi-arid environments,
have nevertheless led to groundwater depletion, computed globally from hydrological models
(e.g., Wada et al., 2010; de Graaf et al., 2017) and volumetric-based calculations (Konikow,
2011). Further, groundwater depletion globally has been shown to contribute to sea-level rise
(e.g., Wada et al., 2016).  However, as recognised in a comprehensive review by Bierkens
and Wada (2019), groundwater depletion is often localised, occurring below the footprint
(200,000 km$^2$) of GRACE as has been well demonstrated by detailed modelling studies in the
California Central Valley (Scanlon et al., 2012a) and North China Plain (Cao et al., 2013).
Projections of the sustainability of groundwater withdrawals under global change are
complicated, in part, by uncertainty in how radiative forcing will affect large-scale, regional
controls on extreme annual precipitation like El Niño Southern Oscillation (Latif and
Keenlyside, 2009). Globally, Reager et al. (2016) show a trend towards enhanced
precipitation on the land under climate change. Given this trend and the observed
intensification of precipitation on land from global warming (Allan et al., 2010; Westra et al.,
2013; Zhang et al., 2013; Myhre et al., 2019), groundwater recharge to many large-scale
aquifer systems may increase under climate change as revealed by the statistical relationships
found in this study between ΔGWS and extreme annual precipitation. The magnitude of this
increase is, however, unlikely to offset the impact of human withdrawals in areas of intensive
abstraction for irrigated agriculture as shown in NW India by Xie et al. (2020).  The
developed set of GRACE-derived ΔGWS time series data for the world's large aquifer
systems provided here offers a consistent, additional benchmark alongside long-term
piezometry to assess not only large-scale climate controls on groundwater replenishment but
also opportunities to enhance groundwater storage through managed aquifer recharge.

## 5   Conclusions

Changes in groundwater storage (ΔGWS) computed from GRACE satellite data continue to rely upon uncertain, uncalibrated estimates of changes in other terrestrial stores of water found in soil, surface water, and snow/ice from global-scale models. The application here of ensemble mean values of three GRACE ΔTWS processing strategies (CSR, JPL-Mascons, GRGS) and five land-surface models (GLDAS 1: CLM, Noah, VIC, Mosaic; GLDAS 2: Noah) is designed to reduce the impact of uncertainty in an individual model or GRACE product on the computation of GRACE-derived ΔGWS. We, nevertheless, identify a few instances where erroneously high or low values of GRACE-derived ΔGWS are computed; these occur primarily in arid and semi-arid environments where uncertainty in the simulation of terrestrial water balances is greatest. Over the period of GRACE observation (2002 to 2016), we show favourable comparisons between GRACE-derived ΔGWS and piezometric observations ($r = 0.62$ to $0.86$) in two contrasting basins (i.e., semi-arid Limpopo Basin, tropical humid Bengal Basin) for which in situ data are available. This study thus contributes to a growing body of research and observations reconciling computed GRACE-derived ΔGWS to ground-based data.

GRACE-derived ΔGWS from 2002 to 2016 for the world's 37 large-scale aquifer systems shows substantial variability as revealed explicitly by 20 potential realisations from GRACE products and LSMs computed here; trends in ensemble mean GRACE-derived ΔGWS are overwhelmingly (87%) non-linear. Linear trends adequately explain variability in GRACE-derived ΔGWS in just 5 aquifer systems for which linear declining trends, indicative of groundwater depletion, are observed in 2 aquifer systems (Arabian, Canning); overall trends for three intensively developed, large-scale aquifer systems (California Central Valley, Ganges-Brahmaputra, North China Plains) are declining but non-linear. This non-linearity in GRACE-derived ΔGWS for the vast majority of the world's large aquifer systems is

inconsistent with previous analyses at the scale of GRACE footprint (~200,000 km$^2$)
asserting global-scale groundwater depletion. Groundwater depletion, more commonly
observed by piezometry, is experienced at scales well below the GRACE footprint and is
likely to be more pervasive than suggested by the presented analysis of large-scale aquifers.
Non-linearity in GRACE-derived ΔGWS arises, in part, from episodic recharge associated
with extreme (>90$^{th}$ percentile) annual precipitation. This episodic replenishment of
groundwater, combined with natural discharges that sustain ecosystem functions and human
withdrawals, produces highly dynamic aquifer systems that complicate assessments of the
sustainability of groundwater withdrawals from large aquifer systems. These findings
highlight, however, potential opportunities for sustaining groundwater withdrawals through
induced recharge from extreme precipitation and managed aquifer recharge.

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

**Acknowledgements**

782 M.S. and R.T. acknowledge support from NERC-ESRC-DFID UPGro '*GroFutures*' (Ref.

NE/M008932/1; www.grofutures.org); R.T. also acknowledges the support of a Royal

Society – Leverhulme Trust Senior Fellowship (Ref. LT170004).

**Data Availability**

Supplementary information is available for this paper as a single PDF file. Data generated

and used in this study can be made available upon request to the corresponding author.

 **Tables and Figures**

 **Table 1.** Identification number, name and general location of the world's 37 large aquifer
 systems as provided in the WHYMAP database (https://www.whymap.org/). Mean climatic
 condition of each of the 37 aquifer systems based on the aridity index is tabulated.

793

| WHYMAP aquifer no. | WHYMAP Aquifer name | Continent | Climate zones based on Aridity index | WHYMAP aquifer no. | WHYMAP Aquifer name | Continent | Climate zones based on Aridity index |
|---|---|---|---|---|---|---|---|
| 1 | Nubian Sandstone Aquifer System | Africa | Hyper-arid | 20 | Maranhao Basin | South America | Humid |
| 2 | Northwestern Sahara Aquifer System | Africa | Arid | 21 | Guarani Aquifer System (Parana Basin) | South America | Humid |
| 3 | Murzuk-Djado Basin | Africa | Hyper-arid | 22 | Arabian Aquifer System | Asia | Arid |
| 4 | Taoudeni-Tanezrouft Basin | Africa | Hyper-arid | 23 | Indus River Basin | Asia | Semi-arid |
| 5 | Senegal-Mauritanian Basin | Africa | Semi-arid | 24 | Ganges-Brahmaputra Basin | Asia | Humid |
| 6 | Iullemmeden-Irhazer Aquifer System | Africa | Arid | 25 | West Siberian Artesian Basin | Asia | Humid |
| 7 | Lake Chad Basin | Africa | Arid | 26 | Tunguss Basin | Asia | Humid |
| 8 | Umm Ruwaba Aquifer (Sudd Basin) | Africa | Semi-arid | 27 | Angara-Lena Basin | Asia | Humid |
| 9 | Ogaden-Juba Basin | Africa | Arid | 28 | Yakut Basin | Asia | Humid |
| 10 | Congo Basin | Africa | Humid | 29 | North China Plains Aquifer System | Asia | Humid |
| 11 | Upper Kalahari-Cuvelai-Zambezi Basin | Africa | Semi-arid | 30 | Song-Liao Plain | Asia | Humid |
| 12 | Lower Kalahari-Stampriet Basin | Africa | Arid | 31 | Tarim Basin | Asia | Arid |
| 13 | Karoo Basin | Africa | Semi-arid | 32 | Paris Basin | Europe | Humid |
| 14 | Northern Great Plains Aquifer | North America | Sub-humid | 33 | East European Aquifer System | Europe | Humid |
| 15 | Cambro-Ordovician Aquifer System | North America | Humid | 34 | North Caucasus Basin | Europe | Semi-arid |
| 16 | California Central Valley Aquifer System | North America | Semi-arid | 35 | Pechora Basin | Europe | Humid |
| 17 | Ogallala Aquifer (High Plains) | North America | Semi-arid | 36 | Great Artesian Basin | Australia | Semi-arid |
| 18 | Atlantic and Gulf Coastal Plains Aquifer | North America | Humid | 37 | Canning Basin | Australia | Arid |
| 19 | Amazon Basin | South America | Humid | | | | |

794

**Table 2.** Variability (expressed as standard deviation) in GRACE-derived estimates of GWS from 20 realisations (3 GRACE-TWS and an ensemble mean of TWS, and 4 LSMs and an ensemble mean of surface water and soil moisture storage, and a snow water storage) and their reported range of uncertainty (% deviation from the ensemble mean) in world's 37 large aquifer systems.

| WHYMAP aquifer no. | WHYMAP Aquifer name | Std. deviation in GRACE-GWS (cm) | Range of uncertainty (%) | WHYMAP aquifer no. | WHYMAP Aquifer name | Std. deviation in GRACE-GWS (cm) | Range of uncertainty (%) |
|---|---|---|---|---|---|---|---|
| 1 | Nubian Sandstone Aquifer System | 1.05 | 83 | 20 | Maranhao Basin | 5.68 | 136 |
| 2 | Northwestern Sahara Aquifer System | 1.29 | 121 | 21 | Guarani Aquifer System (Parana Basin) | 3.37 | 77 |
| 3 | Murzuk-Djado Basin | 1.17 | 189 | 22 | Arabian Aquifer System | 2.01 | 163 |
| 4 | Taoudeni-Tanezrouft Basin | 0.99 | 193 | 23 | Indus River Basin | 3 | 78 |
| 5 | Senegal-Mauritanian Basin | 3.23 | 96 | 24 | Ganges-Brahmaputra Basin | 9.84 | 58 |
| 6 | Iullemmeden-Irhazer Aquifer System | 1.52 | 116 | 25 | West Siberian Artesian Basin | 7.53 | 79 |
| 7 | Lake Chad Basin | 2.23 | 91 | 26 | Tunguss Basin | 7.4 | 103 |
| 8 | Umm Ruwaba Aquifer (Sudd Basin) | 4.95 | 113 | 27 | Angara-Lena Basin | 3.73 | 48 |
| 9 | Ogaden-Juba Basin | 1.52 | 57 | 28 | Yakut Basin | 4.15 | 83 |
| 10 | Congo Basin | 5.09 | 98 | 29 | North China Plains Aquifer System | 3.93 | 77 |
| 11 | Upper Kalahari-Cuvelai-Zambezi Basin | 10.03 | 36 | 30 | Song-Liao Plain | 2.63 | 62 |
| 12 | Lower Kalahari-Stampriet Basin | 1.76 | 106 | 31 | Tarim Basin | 1.37 | 219 |
| 13 | Karoo Basin | 3.06 | 74 | 32 | Paris Basin | 4.06 | 84 |
| 14 | Northern Great Plains Aquifer | 4.18 | 111 | 33 | East European Aquifer System | 5.91 | 75 |
| 15 | Cambro-Ordovician Aquifer System | 4.56 | 44 | 34 | North Caucasus Basin | 4.67 | 66 |
| 16 | California Central Valley Aquifer System | 9.73 | 55 | 35 | Pechora Basin | 8.55 | 94 |
| 17 | Ogallala Aquifer (High Plains) | 4.05 | 104 | 36 | Great Artesian Basin | 2.77 | 69 |
| 18 | Atlantic and Gulf Coastal Plains Aquifer | 2.56 | 193 | 37 | Canning Basin | 5.34 | 57 |
| 19 | Amazon Basin | 10.93 | 58 | | | | |

**Main Figures:**

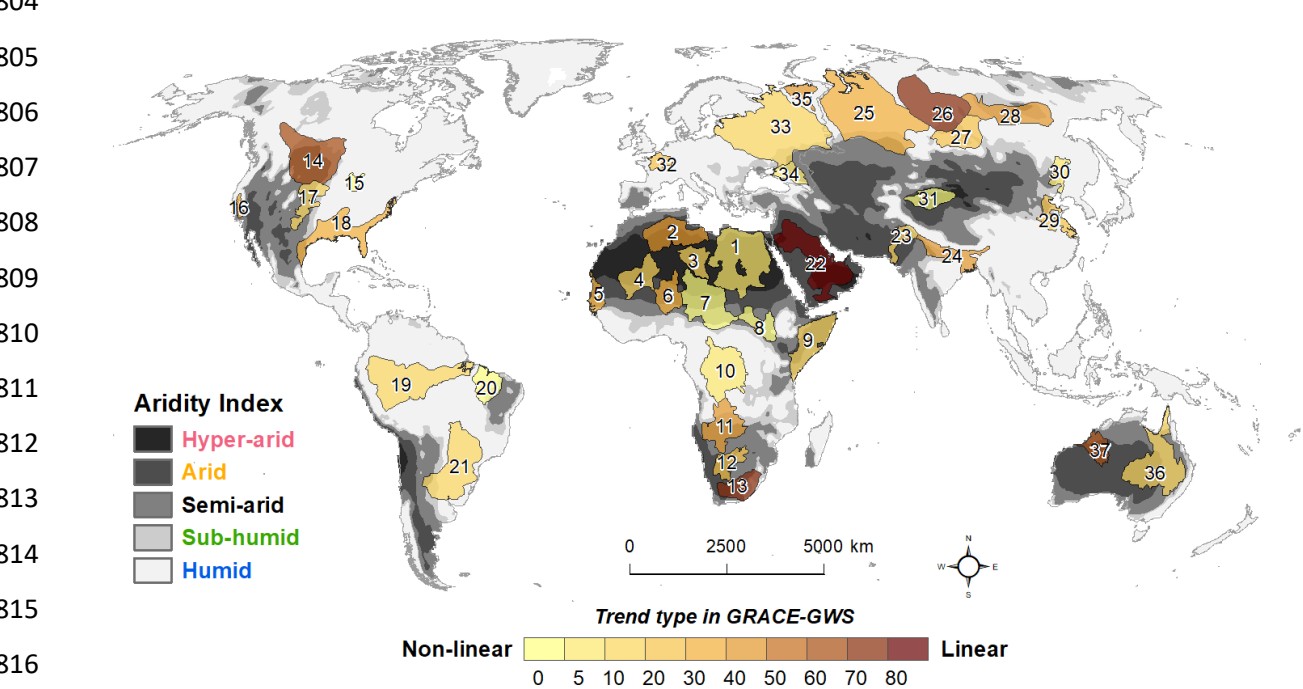

**Fig. 1.** Global map of 37 large aquifer systems from the GIS database of the World-wide Hydrogeological Mapping and Assessment Programme (WHYMAP); names of these aquifer systems are listed in Table 1 and correspond to numbers shown on this map for reference. Grey shading shows the aridity index based on CGIAR's database of the Global Potential Evapo-Transpiration (Global-PET) and Global Aridity Index (https://cgiarcsi.community/); the proportion (as a percentage) of long-term trends in GRACE-derived ΔGWS of these large aquifer systems that is explained by linear trend fitting is shown in colour (i.e. linear trends toward red and non-linear trends toward blue).

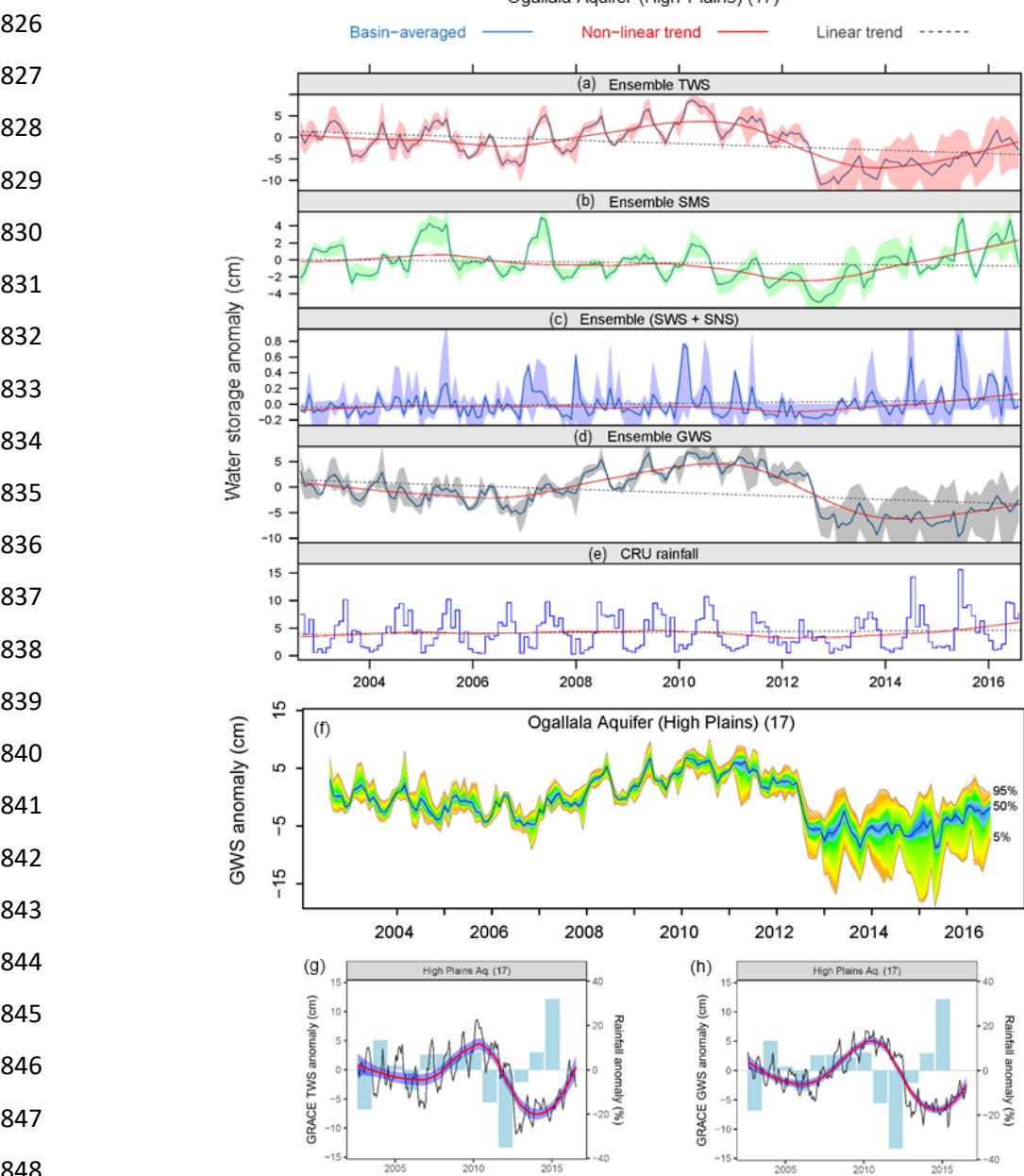

**Fig. 2.** Time-series data of terrestrial water storage anomaly (ΔTWS) from GRACE and individual water stores from GLDAS Land Surface Models (LSMs): (a) Ensemble monthly GRACE ΔTWS from three solutions (CSR, Mascons, GRGS), (b-c) ensemble monthly ΔSMS and ΔSWS + ΔSNS from four GLDAS LSMs (CLM, Noah, VIC, Mosaic), (d) computed monthly ΔGWS and (e) monthly precipitation from August 2002 to July 2016, (f) range of uncertainty in GRACE-derived GWS from 20 realisations, (g) ensemble TWS and annual precipitation, and (h) ensemble GRACE-derived GWS and annual precipitation for the High Plains Aquifer System in the USA (WHYMAP aquifer no. 17). Values in the Y-axis of the top four panels show monthly water-storage anomalies (cm) and the bottom panel shows monthly precipitation (cm). Time-series data (a-e) for the 36 large aquifer systems can be found in supplementary Figs. S1-S36.

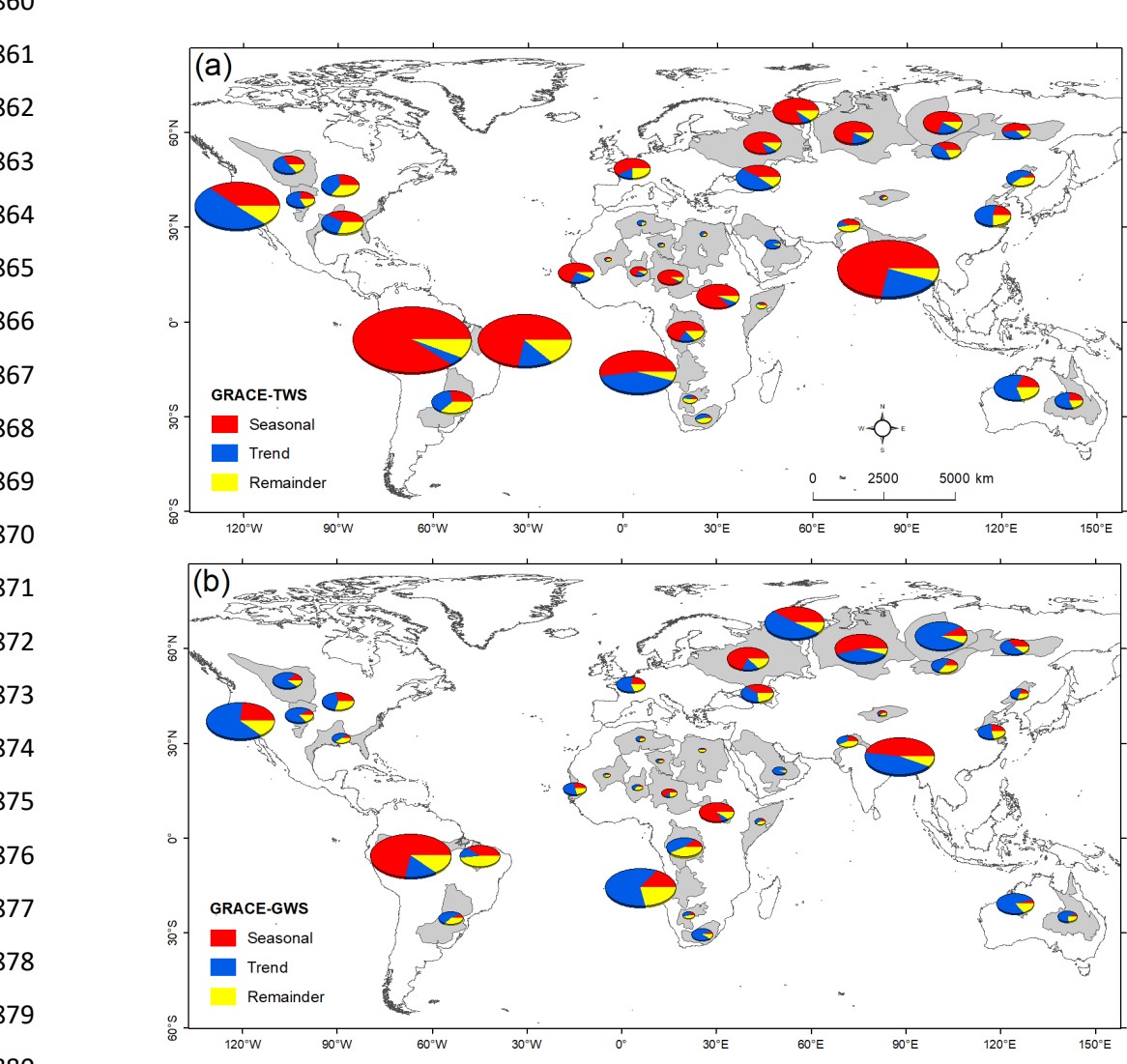

**Fig. 3.** Seasonal-Trend decomposition of (a) GRACE ΔTWS and (b) GRACE ΔGWS time-series data (2002 to 2016) for the world's 37 large aquifer systems using the STL decomposition method; seasonal, trend and remainder or irregular components of time-series data are decomposed and plotted as pie charts that are scaled by the variance of the time series in each aquifer system.

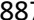

**Fig. 4.** Monthly time-series data (black) of ensemble GRACE ΔTWS for 36 large aquifer systems with a fitted non-linear trend line (Loess smoothing line in thick blue) through the time-series data; GRACE ΔTWS for the remaining large aquifer system (High Plains Aquifer System, (WHYMAP aquifer no. 17) is given in Fig. 2. Shaded area in semi-transparent cyan shows the range of 95% confidence interval of the fitted loess-based non-linear trends; light grey coloured bar diagrams behind the lines on each panel show annual precipitation anomaly (i.e., percentage deviation from the mean precipitation for the period of 1901 to 2016); banner colours indicate the dominant climate of each aquifer based on the mean aridity index shown in the legend on Fig. 1.

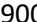

900

901

902

**Fig. 5.** Monthly time-series data (black) of ensemble GRACE ΔGWS for 36 large aquifer
systems with a fitted non-linear trend line (Loess smoothing line in thick blue) through the
time-series data; GRACE ΔGWS for the remaining large aquifer system (High Plains Aquifer
System, (WHYMAP aquifer no. 17) is given in Fig. 2. Shaded area in semi-transparent cyan
shows the range of 95% confidence interval of the fitted loess-based non-linear trends; light
grey coloured bar diagrams behind the lines on each panel show annual precipitation anomaly
(i.e., percentage deviation from the mean precipitation for the period of 1901 to 2016);
banner colours indicate the dominant climate of each aquifer based on the mean aridity index
shown in the legend on Fig. 1.

912

913

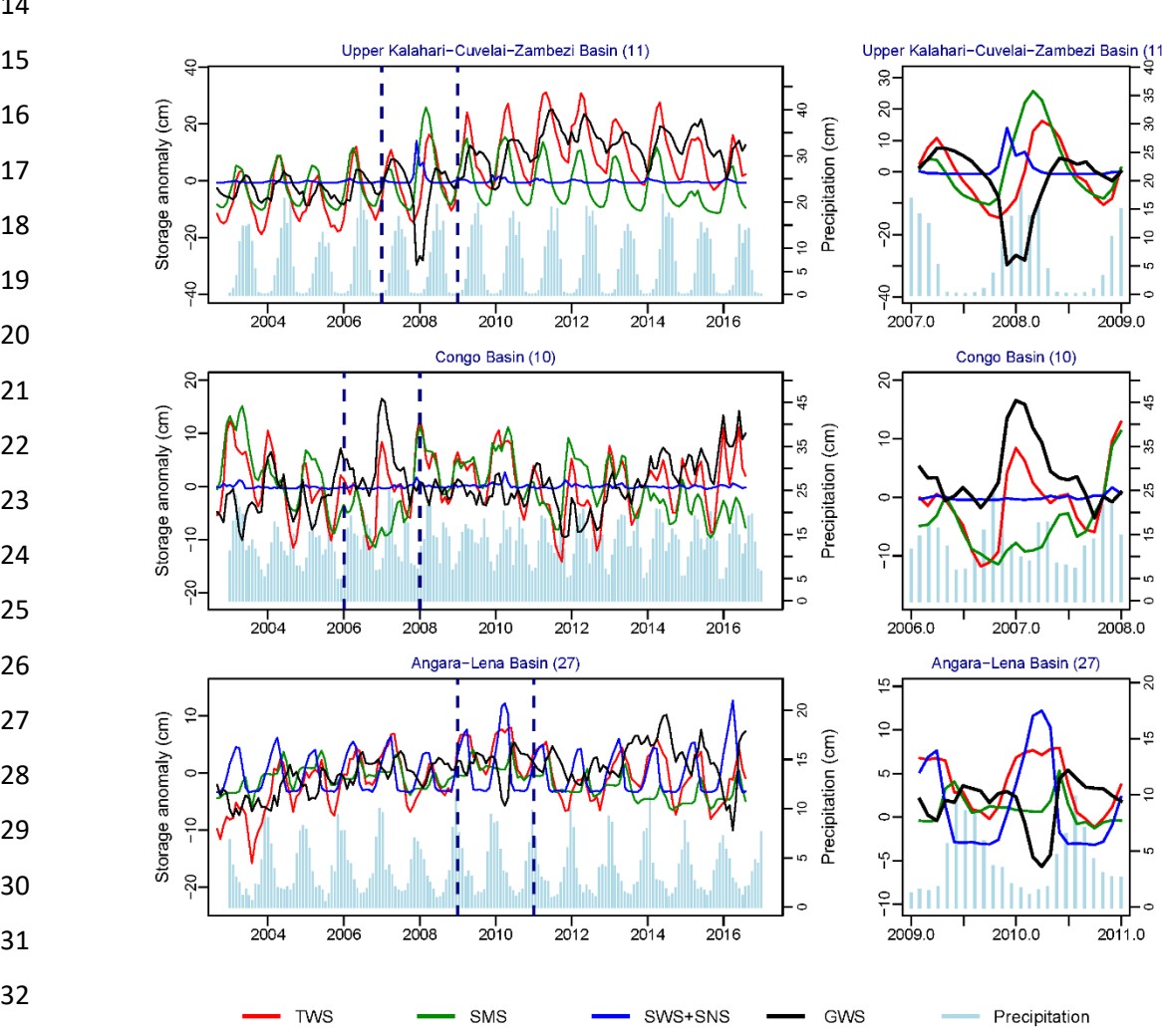

**Fig. 6.** Time series of ensemble mean GRACE ΔTWS (red), GLDAS ΔSMS (green), ΔSWS+ΔSNS (blue) and computed GRACE ΔGWS (black) showing the calculation of anomalously negative or positive values of GRACE ΔGWS that deviate substantially from underlying trends. Three examples include: (a) the Upper Kalahari-Cuvelai-Zambezi Basin (11) under a semi-arid climate; (b) the Congo Basin (10) under a tropical humid climate; and (c) the Angara-Lena Basin (27) under a temperate humid climate; examples from an additional five aquifer systems under semi-arid and arid climates are given in the supplementary material (Fig. S75).

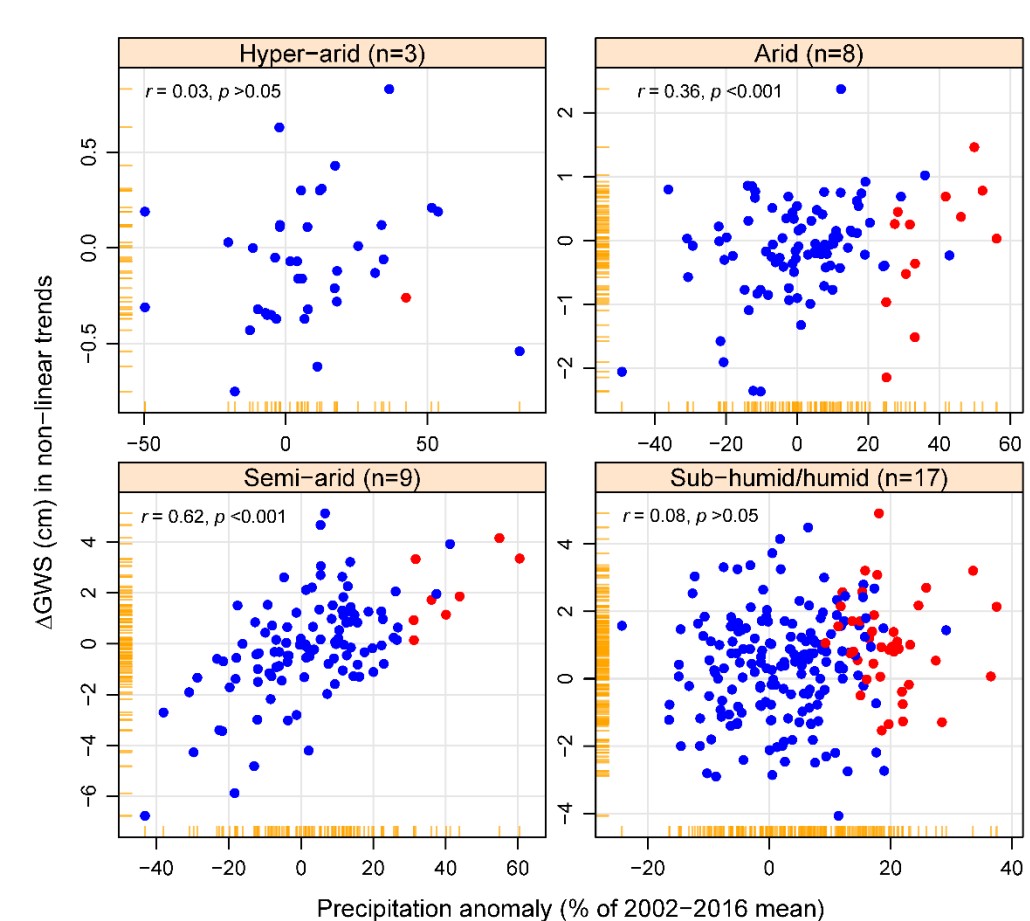

**Fig. 7.** Relationships between precipitation anomaly and annual changes in non-linear trends of GRACE ΔGWS in the 37 large aquifer systems grouped by aridity indices; annual precipitation is calculated based on hydrological year (August to July) for 12 of these aquifer systems and the rest 25 following the calendar year (January to December); the highlighted (red) circles on the scatterplots are the years of statistically extreme (>90th percentile; period: 1901 to 2016) precipitation.

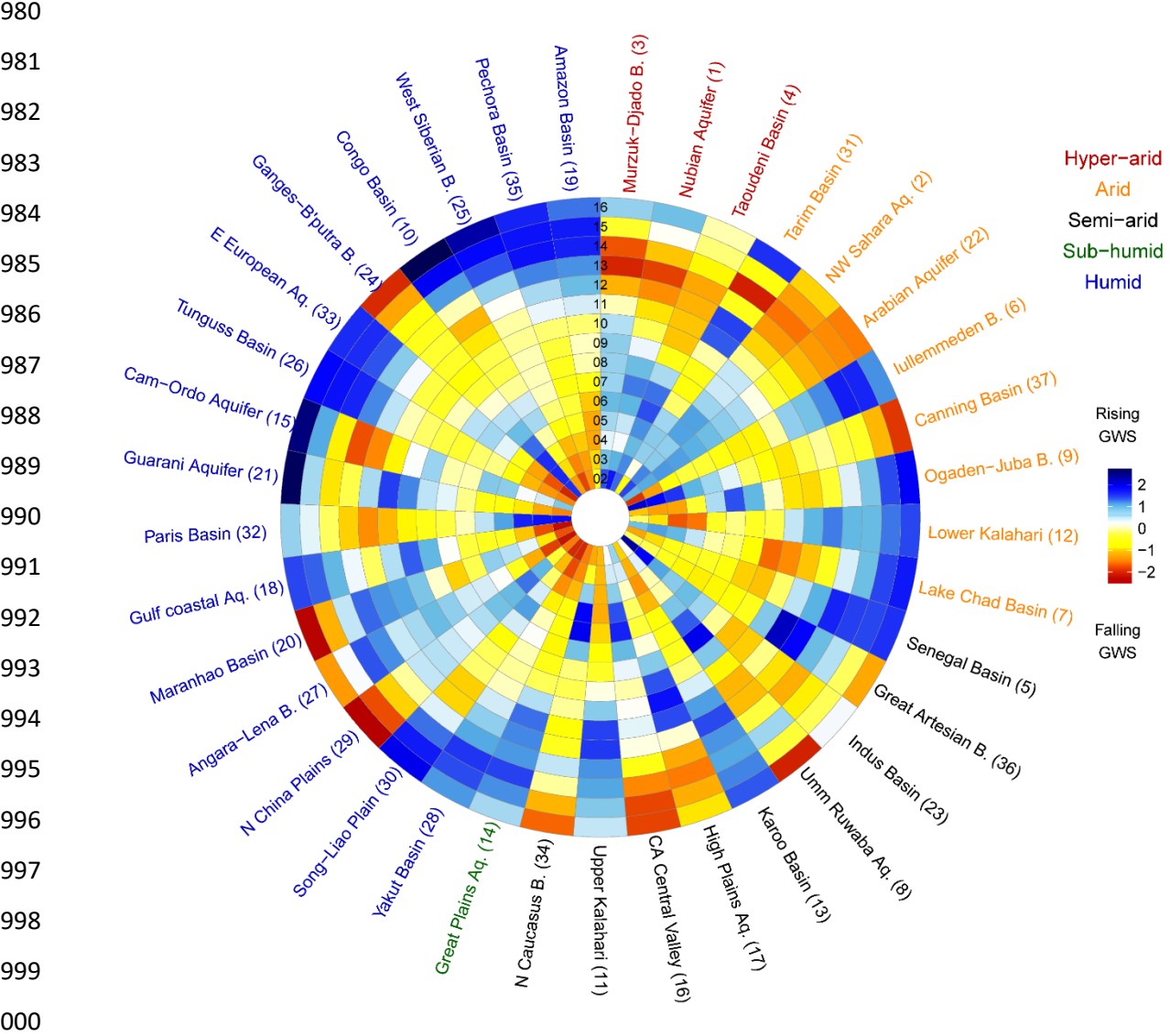

**Fig. 8.** Standardised monthly anomaly of non-linear trends of ensemble mean GRACE ΔGWS for the 37 large aquifer systems from 2002 to 2016. Colours yellow to red indicate progressively declining, short-term trends whereas colours cyan to navy blue indicate rising trends; aquifers are arranged clockwise according to the mean aridity index starting from the hyper-arid climate on top of the circular diagram to progressively humid. Legend colours indicate the climate of each aquifer based on the mean aridity index; time in year (2002 to 2016) is shown from the centre of the circle outwards to the periphery.