# Peer review of "Groundwater storage dynamics in the world's large aquifer systems from GRACE: uncertainty and role of extreme precipitation"

_Earth System Dynamics, 2019_

## Referee Comment (RC1) · Marc Bierkens (Referee) · 2 Jan 2020

**Review of ESD 2019-43 by Mohammad Shamsudduha and Richard G. Taylor**

The authors use the results of three different GRACE-based TWS methods and 4 Land surface models to generate an ensemble of groundwater storage anomalies. These are subsequently analyzed by a non-parametric statistical method to separate seasonal signals from non-linear trends and residuals.

The main message of the paper is that trends in GWS anomalies (ΔGWS), if existing, are non-linear in the vast majority of main aquifer systems and that rainfall anomalies play an important role in explaining these non-linear trends.

I enjoyed reading the paper. I find that it is a well-written with an important message that deserves publication. However, I have a few comments.

**Moderate comments**
1.  I find the lack of reference to estimates based on global hydrological models (GHMS) remarkable. The first spatially distributed global assessment of depletion rates where based on such models and, albeit indirect, should be used in the discussion. They are the basis for the "narratives on global groundwater depletion" that are mentioned in the discussion and the abstract (See https://iopscience.iop.org/article/10.1088/1748-9326/ab1a5f/meta for an overview of these studies). This is the more remarkable, given that the authors do use Land Surface Models (LSMs) to estimate ΔGWS from GRACE ΔTWS.

2.  Regarding the estimation of ΔGWS from GRACE ΔTWS (Equation 1): I am quite doubtful that the surface water storage from integrating LSM runoff on a monthly basis is sufficiently accurate. Even a small basin as the Rhine has a discharge peak routing time of a week, while that of the Amazon amounts to 3 months. Apart from the lack of river routing, GLDAS LSMs do not include the storage and delayed discharge from reservoirs, lakes and inundated floodplains (GHMs do a better job in that respect; see https://agupubs.onlinelibrary.wiley.com/doi/abs/10.1029/2018GL081836 ). This fact may lead to underestimation of ΔSWS and subsequently an overestimation of ΔGWS and its noisiness. Granted, comparison with piezometric data in the Limpopo and the Ganges-Brahmaputra is favourable, but this can be scaled easily by changing specific yield.

3.  The discussion related to the "narrative of global groundwater depletion" needs elaboration:
    *   Not only piezometric studies show that groundwater depletion can be very local; this is also true for model-based estimates of groundwater depletion. See for instance results from Wada et al. 2012 (Figure S5) https://agupubs.onlinelibrary.wiley.com/doi/full/10.1029/2012GL051230 and De Graaf et al 2017 (Figure 11): https://www.sciencedirect.com/science/article/pii/S030917081630656X#fig0011 . This means that at the aquifer scale anomalous rainfall may cause an overall increase in groundwater storage, while groundwater depletion may locally still

persist. Thus, the "narrative of global groundwater depletion" pertains to "groundwater depletion as a global phenomenon".

- The current consensus seems to be that global $\Delta$TWS has been increasing between 1950-1995 by dam building, decreasing from 1995-2005 by groundwater depletion and has been increasing again since then by increased land water storage due to a climate-change induced increase in precipitation: see the review by Wada et al: https://link.springer.com/chapter/10.1007%2F978-3-319-56490-6_7 Yet, at the same time groundwater depletion at the current hotspots has persisted. How do your findings relate to these insights?

4. Line 146: I don't understand the 20 realisations. I would think: 3 GRACE products, 4 LSM estimates of $\Delta$SMS and $\Delta$SWS and one LSM with $\Delta$SNS amount to 3x4x1 = 12 realisations? Or did you combine e.g. $\Delta$SWS from one LSM with the $\Delta$SMS from another? If you did this, this seems to be inconsistent as it would not preserve mass and overestimate the errors due to the LSM corrections.

**Small remarks:**
- The first sentence of the introduction: Doell et al (2012) is only one model-based study providing these numbers. I would advise using less significant numbers based on an overview of estimates by Hanasaki et al (2018): https://www.hydrol-earth-syst-sci.net/22/789/2018/
- Lines 212-215: Trying out different smoothing parameters. I feel that the results of this exercise should be shown, at least in the Supplementary Information (SI). Perhaps report the statistics of the residuals for a number of settings of the smoothing parameters to justify the values chosen.
- On a related note: Looking at some of the plots in the SI I see that residuals are often far from white. In the time series literature this would be seen as a serious model insufficiency. Some discussion on how this would affect results is warranted as well.

---

## Referee Comment (RC2) · Soumendra Bhanja (Referee) · 13 Feb 2020

The authors present a manuscript on GRACE-based terrestrial and groundwater storage changes in 37 major aquifer systems across the globe. I must say, the authors have done a commendable job to compare huge amount of data in all of those major global aquifers. My major comments are provided below:

1. The surface water storage was used from GLDAS estimates of surface runoff. How do the authors comment on surface water storage variations in natural structures such

as, rivers, lakes and artificial structures like dams? I believe, the influence of surface water storage in natural and artificial structures can provide erroneous disaggregation of TWS. The smaller fraction of surface water storage is clearly visible in the figures on comparing the soil moisture and groundwater storages. 2. Lines 304-316: I am not totally agreeing with the arguments provided by the authors. They have not provided sufficient factual evidences in support of these arguments. There can be multiple reasons behind that. Surface water storage in the reservoirs can also play crucial role here, which is not considered in this study. 3. Please include a limitation section mentioning all the limitations in this study. For soil moisture storage, one of the major limitations is that the simulated values are up to 3.4 m at max, soil moisture at deeper layers are not used. This is particularly important in arid, semi-arid regions where vadose zone thickness is much deeper. "Uncertainty is generally higher for aquifers systems located in arid to hyper-arid environments (Table 2, see supplementary Fig. S79)." This observation can be linked with the non-representation of deeper soil moisture. 4. Sections 3.5, 4.2 and elsewhere: In general, while describing extreme precipitation, researchers normally use precipitation per day time-scale. The authors seem not to use the daily precipitation data. Please change the discussion topic to mention precipitation only. 5. Section 3.5: Observing non-significant, low correlation between precipitation and groundwater may indicate human interference. Central valley (16) is a clear exception here. This shows correlation analysis is not properly reflecting the observation. 6. Figure 3: Show the scale of variance. 7. "For example, centennial-scale piezometry in the Ganges-Brahmaputra aquifer system (no. 24) reveals that recent groundwater depletion in NW India traced by GRACE (Fig. 5 and supplementary Fig. S23) follows more than a century of groundwater accumulation through leakage of surface water via a canal network constructed primarily during the 19th century (MacDonald et al., 2016)." Centennial-scale data are not present in the manuscript. Please include them in SI. This is not only from the recharge of canal irrigation, groundwater resources in this area got benefited also from a significant rate of annual rainfall. The present decline is clearly linked to irrigation-linked withdrawal. Please mention these. 8. Figure

8: Continuous rise in GWS observed in several basins including Amazon, where precipitation rates even show declining trends (Figure S18). Please discuss the probable reasons. 9. Line 137: surface runoff or surface water storage ($\Delta$SNS)

---

## Author Comment (AC1) · 12 Mar 2020

**Reviewer's comments are italicised, and our responses are provided in normal fonts.**

**General comments**

*The authors use the results of three different GRACE-based TWS methods and 4 Land surface models to generate an ensemble of groundwater storage anomalies. These are subsequently analyzed by a non-parametric statistical method to separate seasonal signals from non-linear trends and residuals.*

*The main message of the paper is that trends in GWS anomalies (ΔGWS), if existing, are non-linear in the vast majority of main aquifer systems and that rainfall anomalies play an important role in explaining these non-linear trends.*

*I enjoyed reading the paper. I find that it is a well-written with an important message that deserves publication. However, I have a few comments.*

**Moderate comments:**

1. *I find the lack of reference to estimates based on global hydrological models (GHMS) remarkable. The first spatially distributed global assessment of depletion rates where based on such models and, albeit indirect, should be used in the discussion. They are the basis for the "narratives on global groundwater depletion" that are mentioned in the discussion and the abstract (See https://iopscience.iop.org/article/10.1088/1748-9326/ab1a5f/meta for an overview of these studies). This is the more remarkable, given that the authors do use Land Surface Models (LSMs) to estimate ΔGWS from GRACE ΔTWS.*

**Response to reviewer 1 (R1) #1.** Firstly, we thank the reviewer (Professor Bierkens) for his positive comments regarding the manuscript and for providing very constructive suggestions to improve the manuscript. Regarding the lack of reference to global hydrological models (GHMs) in "narratives on global groundwater depletion", we agree that this is a critical omission from the original manuscript, which focused on uncertainty in the estimation of GRACE-derived ΔGWS that is typically reliant on estimates of components of terrestrial storage from LSMs (e.g. Long et al., 2016) including commonly used models from NASA's Global Land Data Assimilation System (GLDAS). The revised manuscript will engage fully and directly with evidence from GHMs in describing narratives of "global groundwater depletion" including the recommended study by Bierkens and Wada (2019) and references therein (e.g. Wada et al., 2010; de Graaf et al., 2017).

2. *Regarding the estimation of ΔGWS from GRACE ΔTWS (Equation 1): I am quite doubtful that the surface water storage from integrating LSM runoff on a monthly basis is sufficiently accurate. Even a small basin as the Rhine has a discharge peak routing time of a week, while that of the Amazon amounts to 3 months. Apart from the lack of river routing, GLDAS LSMs do not include the storage and delayed discharge from reservoirs, lakes and inundated floodplains (GHMs do a better job in that respect; see https://agupubs.onlinelibrary.wiley.com/doi/abs/10.1029/2018GL081836). This fact may lead to underestimation of ΔSWS and subsequently an overestimation of ΔGWS and its noisiness. Granted, comparison with piezometric data in the Limpopo and the*

*Ganges-Brahmaputra is favourable, but this can be scaled easily by changing specific yield.*

**R1 #2.** We agree with the reviewer's concerns regarding the use of GLDAS LSMs to account for surface water storage (ΔSWS) in the estimation of ΔGWS from GRACE, highlighted recently by Scanlon et al. (2019). The original manuscript first notes that most GRACE studies do not account for ΔSWS in the computation of ΔGWS with the assumption that its contribution to ΔTWS is limited. Consistent with previous studies (e.g. Bhanja et al., 2016; Thomas et al., 2017), this study applies time-series simulations of surface runoff from GLDAS LSMs as a proxy for ΔSWS in the absence of global-scale time-series monitoring of surface water storage changes in rivers, lakes, floodplains and reservoirs. Recognition of the problem of routing in the use of GLDAS LSM data for ΔSWS, identified by Reviewer 1, and its implications for the computation of ΔGWS will be made explicit in the revised manuscript. Together with this, we will expand related discussion of the performance of GHMs and LSMs related to GRACE ΔTWS as reviewed recently by Scanlon et al. (2018).

3.  *The discussion related to the "narrative of global groundwater depletion" needs elaboration:*

- *Not only piezometric studies show that groundwater depletion can be very local; this is also true for model-based estimates of groundwater depletion. See for instance results from Wada et al. 2012 (Figure S5) https://agupubs.onlinelibrary.wiley.com/doi/full/10.1029/2012GL051230 and De Graaf et al 2017 (Figure 11): https://www.sciencedirect.com/science/article/pii/S030917081630656X#fig0011*

  *This means that at the aquifer scale anomalous rainfall may cause an overall increase in groundwater storage, while groundwater depletion may locally still persist. Thus, the "narrative of global groundwater depletion" pertains to "groundwater depletion as a global phenomenon".*

**R1 #3a.** We thank Reviewer 1 for their argument that it is not only piezometry but also global-scale modelling that shows that groundwater depletion can be localised so that depletion can occur alongside accumulation in the same (large) aquifer system. The revised manuscript will incorporate evidence from GHMs explicitly in its discussion of the nature of groundwater depletion assessed globally.

- *The current consensus seems to be that global ΔTWS has been increasing between 1950-1995 by dam building, decreasing from 1995-2005 by groundwater depletion and has been increasing again since then by increased land water storage due to a climate-change induced increase in precipitation: see the review by Wada et al: https://link.springer.com/chapter/10.1007%2F978-3-319-56490-6_7. Yet, at the same time groundwater depletion at the current hotspots has persisted. How do your findings relate to these insights?*

**R1 #3b.** We thank R1 for raising an interesting point on the dynamic nature of global ΔTWS due to spatiotemporal variability in anthropogenic influences including climate change on land-water budgets (e.g. irrigation abstraction, construction of dams and reservoirs, trends in precipitation, and land-use change). Our findings on ΔTWS and ΔGWS apply specifically to the period (2002-2016) observed by GRACE. We note from the recommended review by Wada et al. (2017) that recent groundwater-storage depletion has made a net positive contribution to global sea-level rise. Further, as highlighted by R1, Reager et al. (2016) apply GRACE data from 2002 to 2014 to show a trend towards enhanced precipitation on the land under climate change. Given this trend and the observed intensification of precipitation on land under climate change (Allan et al., 2010; Westra et al., 2013; Myhre et al., 2019; Zhang et al., 2013), we may expect that

groundwater recharge to many large-scale aquifer systems will increase under climate change in light of the statistical relationships found in this study between ΔGWS and extreme precipitation. We propose to update the discussion on this specific point in the revised manuscript.

*4. Line 146: I don't understand the 20 realisations. I would think: 3 GRACE products, 4 LSM estimates of ΔSMS and ΔSWS and one LSM with ΔSNS amount to 3x4x1 = 12 realisations? Or did you combine e.g. ΔSWS from one LSM with the ΔSMS from another? If you did this, this seems to be inconsistent as it would not preserve mass and overestimate the errors due to the LSM corrections.*

**R1 #4.** We thank Reviewer 1 for this query and will revise our explanation from where the 20 realisations derive. On lines 142-145 of the original manuscript, we write, "we apply 3 gridded GRACE products (CSR, JPL-Mascons, GRGS) and an ensemble mean of ΔTWS and individual storage component of ΔSMS and ΔSWS from 4 Land Surface Models (LSMs: CLM, Noah, VIC, Mosaic), and a single ΔSNS from Noah model (GLDAS version 2.1)." The breakdown of 20 realisations is given below with 12 realisations being the primary products, whereas the remaining 8 realisations derive from a combination of GRACE ΔTWS and different LSMs to demonstrate the range of uncertainty in the estimation of ΔGWS using GRACE-derived ΔTWS and GLDAS LSMs.

- 3 GRACE TWS x 4 LSMs (SWS, SMS) x 1 LSM (SNS) = 12 realisations
- CSR GRACE TWS x mean LSMs (SWS, SMS) x 1 LSM (SNS) = 1 realisation
- JPL GRACE TWS x mean LSMs (SWS, SMS) x 1 LSM (SNS) = 1 realisation
- GRGS GRACE TWS x mean LSMs (SWS, SMS) x 1 LSM (SNS) = 1 realisation
- Mean GRACE TWS x 4 LSMs (SWS, SMS) x 1 LSM (SNS) = 4 realisation
- Mean GRACE TWS x mean LSMs (SWS, SMS) x 1 LSM (SNS) = 1 realisation

**Small remarks:**

- *The first sentence of the introduction: Doell et al (2012) is only one model-based study providing these numbers. I would advise using less significant numbers based on an overview of estimates by Hanasaki et al (2018): https://www.hydrol-earthsyst-sci.net/22/789/2018/*

**R1 #5.** We appreciate Reviewer 1's suggestion and will incorporate this recommended study by Hanasaki et al. (2018) in the revised manuscript.

- *Lines 212-215: Trying out different smoothing parameters. I feel that the results of this exercise should be shown, at least in the Supplementary Information (SI). Perhaps report the statistics of the residuals for a number of settings of the smoothing parameters to justify the values chosen.*

**R1 #6.** The original analysis evaluated the effect of seasonal and trend smoothing windows in applying the STL (Seasonal-Trend Decomposition using Loess) decomposition method. In the revised manuscript, we will expand discussion in the Methodology of the sensitivity analysis that applied smoothing windows at various lengths; we will also include a new figure to the supplementary information in addition to the current STL figure in Fig. S37.

- *On a related note: Looking at some of the plots in the SI I see that residuals are often far from white. In the time series literature this would be seen as a serious model insufficiency. Some discussion on how this would affect results is warranted as well.*

**R1 #7.** We thank Reviewer 1 for spotting the fact that some lines (e.g. uncertainty envelop around the mean of time-series records) are cut-off by figure margins. In the revised

supplementary information, we will reproduce all the time-series plots (from Figs S1 to S36) with a full range of values.

**References**

Allan, R. P., Soden, B. J., John, V. O., Ingram, W., and Good, P.: Current changes in tropical precipitation, Environmental Research Letters, 5, 025205, 10.1088/1748-9326/5/2/025205, 2010.

Bhanja, S. N., Mukherjee, A., Saha, D., Velicogna, I., and Famiglietti, J. S.: Validation of GRACE based groundwater storage anomaly using in-situ groundwater level measurements in India, Journal of Hydrology, 543, 729-738, 2016.

Bierkens, M. F. P., and Wada, Y.: Non-renewable groundwater use and groundwater depletion: a review, Environ. Res. Lett., 14, 063002, 10.1088/1748-9326/ab1a5f, 2019.

de Graaf, I. E. M., van Beek, L. P. H., Gleeson, T., Moosdorf, N., Schmitz, O., Sutanudjaja, E. H., and Bierkens, M. F. P.: A global-scale two layer transient groundwater model: development and application to groundwater depletion, Adv. Water Resour., 102, 53–67, 2017.

Hanasaki, N., Yoshikawa, S., Pokhrel, Y., and Kanae, S.: A global hydrological simulation to specify the sources of water used by humans, Hydrol. Earth Syst. Sci., 22, 789–817, 2018.

Long, D., Chen, X., Scanlon, B. R., Wada, Y., Hong, Y., Singh, V. P., Chen, Y., Wang, C., Han, Z., and Yang, W.: Have GRACE satellites overestimated groundwater depletion in the Northwest India Aquifer?, Scientific Reports, 6, 24398, doi:10.1038/srep24398, 2016.

Myhre, G., Alterskjær, K., Stjern, C. W., Hodnebrog, Ø., Marelle, L., Samset, B. H., Sillmann, J., Schaller, N., Fischer, E., Schulz, M., and Stohl, A.: Frequency of extreme precipitation increases extensively with event rareness under global warming, Scientific Reports, 9, 16063, 10.1038/s41598-019-52277-4, 2019.

Reager, J. T., Gardner, A. S., Famiglietti, J. S., Wiese, D. N., Eicker, A., and Lo, M.-H.: A decade of sea level rise slowed by climate-driven hydrology, Science, 351, 699-703, 2016.

Scanlon, B. R., Zhang, Z., Save, H., Sun, A. Y., Müller Schmied, H., van Beek, L. P. H., Wiese, D. N., Wada, Y., Long, D., Reedy, R. C., Longuevergne, L., Döll, P., and Bierkens, M. F. P.: Global models underestimate large decadal declining and rising water storage trends relative to GRACE satellite data, PNAS, 115 1080-1089, doi:10.1073/pnas.1704665115, 2018.

Scanlon, B. R., Zhang, Z., Rateb, A., Sun, A., Wiese, D., Save, H., Beaudoing, H., Lo, M. H., Müller-Schmied, H., Döll, P., van Beek, R., Swenson, S., Lawrence, D., Croteau, M., and Reedy, R. C.: Tracking seasonal fluctuations in land water storage using global models and GRACE satellites, Geophysical Research Letters, 46, 5254–5264, 10.1029/2018GL081836, 2019.

Thomas, B. F., Caineta, J., and Nanteza, J.: Global assessment of groundwater sustainability based on storage anomalies, Geophysical Research Letters, 44, 11445-11455, doi:10.1002/2017GL076005, 2017.

Wada, Y., Beek, L. P. H. v., Kempen, C. M. v., Reckman, J. W. T. M., S. Vasak, and Bierkens, M. F. P.: Global depletion of groundwater resources, Geophys. Res. Lett., 37, L20402, 10.1029/2010GL044571, 2010.

Wada, Y., Reager, J. T., Chao, B. F., Wang, J., Lo, M.-H., Song, C., Li, Y., and Gardner, A. S.: Recent Changes in Land Water Storage and its Contribution to Sea Level Variations, Surv Geophys, 38, 131–152, 10.1007/s10712-016-9399-6, 2017.

Westra, S., Alexander, L. V., and Zwiers, F. W.: Global Increasing Trends in Annual Maximum Daily Precipitation, J. Climate, 26, 3904–3918, 10.1175/JCLI-D-12-00502.1, 2013.

Zhang, X., Wan, H., Zwiers, F. W., Hegerl, G. C., and Min, S.-K.: Attributing intensification of precipitation extremes to human influence, Geophysical Research Letters, 40, 5252–5257, 10.1002/grl.51010, 2013.

---

## Author Comment (AC2) · 12 Mar 2020

**Reviewer's comments are italicised, and our responses are provided in normal fonts.**

**General comments:**

*The authors present a manuscript on GRACE-based terrestrial and groundwater storage changes in 37 major aquifer systems across the globe. I must say, the authors have done a commendable job to compare huge amount of data in all of those major global aquifers.*

**My major comments are provided below:**

1. *The surface water storage was used from GLDAS estimates of surface runoff. How do the authors comment on surface water storage variations in natural structures such as, rivers, lakes and artificial structures like dams? I believe, the influence of surface water storage in natural and artificial structures can provide erroneous disaggregation of TWS. The smaller fraction of surface water storage is clearly visible in the figures on comparing the soil moisture and groundwater storages.*

**R2 #1.** We thank Reviewer 2 (R2, Dr. Bhanja) for his positive comments and providing valuable suggestions to improve the manuscript. As with R1, we agree with R2 that the representation of surface water storage (ΔSWS) by NASA's GLDAS (Global Land Data Assimilation System) models (i.e. CLM, Noah, VIC, Mosaic) is uncertain and limited. As with a recent study (Getirana et al., 2017), we note that the vast majority of studies estimating GRACE-derived ΔGWS disregard ΔSWS, assuming its contribution to ΔTWS variability is small (e.g. Long et al., 2016) in contrast to other studies (e.g. Shamsudduha et al., 2012) that show that contribution to ΔTWS from observed ΔSWS is substantial (22%). However, more precise estimates of the impacts of ΔSWS on ΔTWS globally and its spatial variability are unknown due to lack of global-scale observations of surface water storage including anthropogenic structures such as dams and reservoirs. Similar to the approaches taken in recent studies (e.g. Bhanja et al., 2016; Thomas et al., 2017), we apply surface runoff as a reasonable proxy for ΔSWS derived from GLDAS LSMs. We recognise differences exist in surface runoff estimates simulated by GLDAS LSMs as noted previously in inter-comparison studies (e.g. Scanlon et al., 2018; Scanlon et al., 2019; Zaitchik et al., 2010). We contend, however, that dismissing ΔSWS entirely in the estimation of ΔGWS from GRACE does not serve to reduce uncertainty.

2. *Lines 304-316: I am not totally agreeing with the arguments provided by the authors. They have not provided sufficient factual evidences in support of these arguments. There can be multiple reasons behind that. Surface water storage in the reservoirs can also play crucial role here, which is not considered in this study.*

**R2 #2.** On lines 304 to 316 in the original manuscript, we argue that abrupt rises or falls in calculated ΔGWS can result from simple arithmetic operations of numeric values, given the uncertainty that exists in estimates of water storage from GLDAS LSMs expressed as anomalies. We agree with R2 that uncertainty in the estimation of ΔSWS may be one of the causes of these abrupt rises or falls. Because ΔGWS is calculated from GRACE-derived ΔTWS by subtracting storage anomalies from other terrestrial water components such as soil moisture storage (ΔSMS), surface water storage (ΔSWS) and snow water storage (ΔSNS), 'mathematical artefacts' in the calculation of groundwater storage change (ΔGWS)

from GRACE and GLDAS datasets can occur where over/underestimation in individual components can collectively lead to abrupt falls/rises relative to GRACE ΔTWS as per equation 1 in the original manuscript.

3. *Please include a limitation section mentioning all the limitations in this study. For soil moisture storage, one of the major limitations is that the simulated values are up to 3.4 m at max, soil moisture at deeper layers are not used. This is particularly important in arid, semi-arid regions where vadose zone thickness is much deeper. "Uncertainty is generally higher for aquifers systems located in arid to hyper-arid environments (Table 2, see supplementary Fig. S79)." This observation can be linked with the non-representation of deeper soil moisture.*

**R2 #3.** We agree with R2 that substantial variability and uncertainty exists in simulated soil moisture storage by GLDAS land surface models. R2 is correct in noting that the number of layers and depth of soil horizons in the four LSMs differ with a maximum depth of 3.5 m in Mosaic. We agree that the depth to the deepest part of the unsaturated zone and soils in semi-arid and arid environments could be well below the maximum depth of soil layers considered in these models. For example, the thickness of unsaturated zone in the Southern High Plains in the US can range from 0 to 134 m with a median thickness of 37 m (Scanlon et al., 2009). We will nevertheless expand discussion of uncertainty in the representation of ΔSMS by GLDAS LSMs making specific reference to their consequences for soils in semi-arid and arid environments.

4. *Sections 3.5, 4.2 and elsewhere: In general, while describing extreme precipitation, researchers normally use precipitation per day time-scale. The authors seem not to use the daily precipitation data. Please change the discussion topic to mention precipitation only.*

**R2 #4.** We agree with R2 that precipitation intensity is most commonly discussed in relation to daily precipitation but this is not exclusive. Statistically, extreme precipitation can also refer to annual, seasonal, and monthly precipitation. In the manuscript, we define statistically extreme precipitation (i.e. 90th percentiles) on a monthly timestep over the period of 1901 to 2016 (116 years) consistent with the monthly timestep in the employed GRACE and GLDAS time-series datasets.

5. *Section 3.5: Observing non-significant, low correlation between precipitation and groundwater may indicate human interference. Central valley (16) is a clear exception here. This shows correlation analysis is not properly reflecting the observation.*

**R2 #5.** We agree with the R2 that the low correlation between precipitation and groundwater storage may indicate human interference (i.e. groundwater pumping) masking natural variability that may be caused by climate (i.e. precipitation variations). In the revised manuscript, we will expand discussion of potential factors overprinting natural variabilities in ΔGWS such as groundwater abstraction.

6. *Figure 3: Show the scale of variance.*

**R2 #6.** The variance in GRACE-derived TWS time-series records for all 37 large aquifer systems are presented in the Supplementary Table S1.

7. *"For example, centennial-scale piezometry in the Ganges-Brahmaputra aquifer system (no. 24) reveals that recent groundwater depletion in NW India traced by GRACE (Fig. 5 and supplementary Fig. S23) follows more than a century of groundwater accumulation through leakage of surface water via a canal network constructed primarily during the 19th century (MacDonald et al., 2016)." Centennial-scale data are not present in the manuscript. Please include them in SI. This is not only from the recharge of canal irrigation, groundwater resources in this area got benefited also from a significant rate of*

*annual rainfall. The present decline is clearly linked to irrigation-linked withdrawal. Please mention these.*

**R2 #7.** We thank R2 for their comments and suggestions regarding centennial-scale changes in groundwater storage in the Ganges-Brahmaputra Basin that contrast with short-term (2002-2016) declining trends in ΔGWS revealed by GRACE. We agree that recent declining trends result from intensive groundwater-fed irrigation in the region. The centennial-scale groundwater levels (1900 to 2010) are clearly shown in Figure 3b of the *Nature Geoscience* letter by MacDonald et al. (2016) and we will reproduce this figure, as proposed, in the revised Supplementary Information.

8.  *Figure 8: Continuous rise in GWS observed in several basins including Amazon, where precipitation rates even show declining trends (Figure S18). Please discuss the probable reasons.*

**R2 #8.** We similarly note this interesting observation made by R2. Rising trends in GRACE-derived ΔGWS time series follows a similar trend in ΔTWS over the Amazon Basin that has been reported by Scanlon et al. (2018) at 41 to 44 km$^3$/year (period 2002 to 2014) and Rodell et al. (2018) at 51.9 ± 9.4 Gt/year (period 2002 to 2016). The magnitude of this rising trend in ΔTWS is explained by both the size of the region and the intensity of the Amazon water cycle (Chaudhari et al., 2019). Furthermore, Rodell et al. (2018) argue that large dam construction in southern Brazil and filling of reservoirs contributed to the rising trend in GRACE ΔTWS. We note that a slightly decreasing trend in soil moisture storage (ΔSMS) might be contributing to a positive change in ΔGWS over the Amazon Basin.

9.  *Line 137: surface runoff or surface water storage (SNS).*

**R2 #9.** Thanks to R2 for pointing out this typo. We agree that it should read SWS, not SNS and will be corrected in the revised manuscript.

**References**

Bhanja, S. N., Mukherjee, A., Saha, D., Velicogna, I., and Famiglietti, J. S.: Validation of GRACE based groundwater storage anomaly using in-situ groundwater level measurements in India, Journal of Hydrology, 543, 729-738, 2016.

Chaudhari, S., Pokhrel, Y., Moran, E., and Miguez-Macho, G.: Multi-decadal hydrologic change and variability in the Amazon River basin: understanding terrestrial water storage variations and drought characteristics, Hydrol. Earth Syst. Sci., 23, 2841–2862, 10.5194/hess-23-2841-2019, 2019.

Getirana, A., Kumar, S., Girotto, M., and Rodell, M.: Rivers and Floodplains as Key Components of Global Terrestrial Water Storage Variability, Geophysical Research Letters, 44, 10359-10368, 2017.

Long, D., Chen, X., Scanlon, B. R., Wada, Y., Hong, Y., Singh, V. P., Chen, Y., Wang, C., Han, Z., and Yang, W.: Have GRACE satellites overestimated groundwater depletion in the Northwest India Aquifer?, Scientific Reports, 6, 24398, doi:10.1038/srep24398, 2016.

MacDonald, A. M., Bonsor, H. C., Ahmed, K. M., Burgess, W. G., Basharat, M., Calow, R. C., Dixit, A., Foster, S. S. D., Gopal, K., Lapworth, D. J., Lark, R. M., Moench, M., Mukherjee, A., Rao, M. S., Shamsudduha, M., Smith, L., Taylor, R. G., Tucker, J., van Steenbergen, F., and Yadav, S. K.: Groundwater quality and depletion in the Indo-Gangetic Basin mapped from in situ observations, Nature Geoscience, 9, 762-766, 2016.

Rodell, M., Famiglietti, J. S., Wiese, D. N., Reager, J. T., Beaudoing, H. K., Landerer, F. W., and Lo, M. H.: Emerging trends in global freshwater availability, Nature, 557, 651-659, 10.1038/s41586-018-0123-1, 2018.

Scanlon, B. R., Stonestrom, D. A., Reedy, R. C., Leaney, F. W., Gates, J., and Cresswell, R. G.: Inventories and mobilization of unsaturated zone sulfate, fluoride, and chloride related to land use change in semiarid regions, southwestern United States and Australia, Water Resour. Res., 45, W00A18, 10.1029/2008WR006963, 2009.

Scanlon, B. R., Zhang, Z., Save, H., Sun, A. Y., Müller Schmied, H., van Beek, L. P. H., Wiese, D. N., Wada, Y., Long, D., Reedy, R. C., Longuevergne, L., Döll, P., and Bierkens, M. F. P.: Global models underestimate large decadal declining and rising water storage trends relative to GRACE satellite data, PNAS, 115 1080-1089, doi:10.1073/pnas.1704665115, 2018.

Scanlon, B. R., Zhang, Z., Rateb, A., Sun, A., Wiese, D., Save, H., Beaudoing, H., Lo, M. H., Müller-Schmied, H., Döll, P., van Beek, R., Swenson, S., Lawrence, D., Croteau, M., and Reedy, R. C.: Tracking seasonal fluctuations in land water storage using global models and GRACE satellites, Geophysical Research Letters, 46, 5254–5264, 10.1029/2018GL081836, 2019.

Shamsudduha, M., Taylor, R. G., and Longuevergne, L.: Monitoring groundwater storage changes in the highly seasonal humid tropics: validation of GRACE measurements in the Bengal Basin, Water Resour. Res., 48, W02508, doi:10.1029/2011WR010993, 2012.

Thomas, B. F., Caineta, J., and Nanteza, J.: Global assessment of groundwater sustainability based on storage anomalies, Geophysical Research Letters, 44, 11445-11455, doi:10.1002/2017GL076005, 2017.

Zaitchik, B. F., Rodell, M., and Olivera, F.: Evaluation of the Global Land Data Assimilation System using global river discharge data and a source-to-sink routing scheme, Water Resour. Res. , 46, W06507, 10.1029/2009WR007811, 2010.